# The impact of new urbanization on "quantity increase and quality improvement" of urban green innovation

Liang Fang[ID][1¤a]*, Chengxiang Li[2¤b], Chen Jin[ID][1¤a]

1  School of Economics and Management, Huangshan University, Huangshan City, Anhui Province, China,
2  School of Management, Anhui Science and Technology University, Bengbu City, Anhui Province, China

¤a Current address: No. 39, Xihai Road, Tunxi District, Huangshan University, Huangshan City, Anhui Province, China
¤b Current address: Anhui Science and Technology University, Bengbu City, Anhui Province, China
* 101028@hsu.edu.cn

## Abstract

With the rise of new urbanization (NEU) as a major national strategy, a proposition with both theoretical and practical significance has been highlighted. Whether the implementation of this strategy can substantially enhance the quantity and quality of urban green innovation (GI) remains a focus of ongoing attention from the government and academia. Based on panel data from 284 cities from 2011 to 2022, this study constructs two-way fixed-effects models and multivariate moderation models, and empirically analyzes the effect of NEU on the "quantity increase" and "quality improvement" of urban GI. The study finds that the coefficients for NEU's impact on "quantity increase" and "quality improvement" of urban GI are statistically significant at the 0.01 level, with positive signs indicating enhancement effects. The moderating effect indicates that the coupling interaction between digital inclusive finance and various elements (government technology support, informatization, regional economic development, energy consumption) constitutes a gain mechanism, effectively strengthening the marginal effect of increases in the quantity and quality of urban GI. Threshold effect analysis indicates that the government's green support exerts a threshold effect on NEU's influence on urban GI. The paper not only deepens the understanding of the NEU dividend release mechanism but also provides operational policy implications for coordinating the two key agendas of NEU construction and GI development.

## 1.  Introduction

China's rapid urbanization in recent years has triggered a profound reconstruction of social space, and the process of the citizenization of a large number of rural residents has accelerated the rise of urban agglomerations. These emerging urban

**Data availability statement:** The data are held in the public repository Zenodo (DOI: 10.5281/zenodo.17547697).

**Funding:** This research is funded by the following grants: Humanities and Social Sciences Research Project of the Ministry of Education of China (20YJCZH028 to LF), Anhui Province Outstanding Talents Cultivation Project of Universities (gxbjZD2022062 to LF), Huangshan University Scientific Research Innovation Team Project (2021XCXTDPY04 to LF).

**Competing interests:** The authors have declared that no competing interests exist.

agglomerations, combining high-end industries and innovative resources, have become a key growth pole and power source supporting the sustained, rapid growth of China's economy [1]. The problems of resource overconsumption and environmental pollution that have accumulated during traditional urbanization are particularly prominent. The path dependence of high input and high emission brought by urbanization has become a key bottleneck hindering the high-quality development of the economy and society [2]. Due to path dependence in traditional, low-level, labor-intensive industries, the demand for technological innovation in some cities has shown a relatively low trend over an extended period. As China's response, the new urbanization (NEU) strategy does not prioritize city size and population. However, it is committed to building a new model of modern urban development with complete functions, high-quality services, and ecological friendliness [3]. "Based on the policy guidance of the '2025 Government Work Report of the State Council', China has established a strategic policy of innovation-driven high-quality development of NEU. This policy aims to consolidate the foundation of green sustainable development by optimizing resource allocation and accelerating industrial low-carbon transformation. However, amid fierce global competition in the green industry and profound energy structure adjustments, how to leverage NEU's opportunities to cultivate new, high-quality productivity and promote economic growth to achieve a substantial green recovery and long-term, healthy development has become an urgent issue of the times [4]. The key to addressing the dual pressures of the economy and the environment is to lead green innovation (GI) with NEU, promote the low-carbon transformation of the social economy, and achieve a leap in green technology. In view of this, this paper aims to clarify whether the NEU has injected new kinetic energy into urban green technology innovation and to explore its internal logic of maximizing economic, social and ecological benefits, to provide theoretical support for exploring the feasible path of urban sustainable development.

In the existing literature, NEU has been shown to have both environmental improvement effects and ecological remodeling functions: it not only effectively reduces pollution intensity and optimizes energy consumption structure, but also profoundly alters regional ecological balance and development sustainability [5]. By guiding production factors to gather in cities and towns, the NEU has played a dual role in industrial upgrading, regional integration, and the release of scale economies. This series of chain reactions together constitutes a solid foundation and strong thrust for urban economic development [6]. Academic research has explored the correlation between urbanization and the development of information technology, as well as the resulting digital divide. Wang Z et al. (2019), from the perspective of environmental governance, proposed that NEU has achieved indirect suppression of carbon dioxide emissions by blocking the ' rebound effect ' of energy-saving technologies and promoting environmental technology progress. Their core finding is that the mismatch between energy conservation and environmental technology development is the root cause of the significant threshold effect of the two on carbon emissions [7]. There are also studies in the academic community that focus on how the urbanization model affects the stage evolution of the innovation process and triggers inter-regional

spillover effects. Tripathi S et al. (2022) noted that although urbanization can effectively promote the progressive development of regional innovation activities at all stages, there are significant differences in the marginal effects of different urbanization models across different links, such as knowledge creation, knowledge implementation, and innovation production [8]. In exploring the environmental effects of NEU, Zhang Ran et al. (2023) revealed its internal mechanism: the strategy provides key financial support for green technology innovation by alleviating the financial resource bottleneck faced by enterprises, thereby establishing a complete transmission path from institutional supply to environmental performance improvement [9]. In the field of urbanization innovation policy research, scholars are committed to analyzing how policy intervention reshapes regional innovation capabilities and to deeply analyzing its internal heterogeneity, logic, and mechanisms of action. Li S et al. (2023) pointed out that the NEU pilot policy has injected a key endogenous driving force into urban innovation by consolidating the foundation of human capital. It is worth noting that this empowerment effect is not homogeneous across the whole region, but shows significant differentiation along the dimensions of urban level (first-line and second-line) and geographical location (west), and a stronger marginal contribution in these regions [10]. In view of the spatial interaction between urbanization and innovation, Chen J et al. (2020) revealed its potential dialectics. Urbanization can significantly drive a leap in innovation energy levels in the region; it may also create spatial inhibition and crowding-out effects on the innovation capacity of surrounding areas through one-way agglomeration of resources [11]. In the literature on the interaction between urban GI and NEU, the studies by Ma C et al. (2022) and Razzaq A et al. (2022) are quite representative. They go beyond a single dimension, focusing on the analysis of the mutual feedback mechanism between the two systems and its deep impact on the ecological environment's carrying capacity and the resource metabolism process [12,13]. A review of the existing literature indicates that most researchers agree that NEU can effectively promote urban GI. In contrast, a few argue that NEU may be a potential obstacle to urban GI. Zhao et al. (2025) proposed that NEU faces resource and environmental constraints throughout the period of expedited development, which may limit the space for and the impact of urban GI. Especially in areas with poor resource endowments, the role of NEU in promoting urban GI is limited [14].

The existing related research does not systematically analyze the quantity and quality of GI as independent explanatory variables, nor does it clearly distinguish the driving mechanisms between "quantity increase" and "quality improvement". The "quantity increase" reflects the activity and popularity of GI, and the "quality improvement" reflects the depth and level of GI. Dividing GI into "quantity increase" and "quality improvement" can more comprehensively evaluate the impact of NEU on GI and avoid the one-sidedness of policy-effect evaluation, providing a more accurate and pertinent basis for policy formulation and also guiding the development direction of GI practice. In existing research, there is also a lack of systematic analysis based on multi-theoretical perspectives, such as urban development theory, GI theory, digital finance theory, and agglomeration effects. Most of the moderating factors, such as government support and digital inclusive finance, are assumed to be linear and lack nonlinear tests such as threshold effects. This limits our comprehensive understanding of the GI effects of NEU and weakens the accuracy of policy formulation.

The contributions of this paper are: (1)Agglomeration economic theory and growth pole theory mostly focus on the linear growth of quantity, and the regional development theory also mainly discusses innovation in a broad sense. Unlike the single-dimensional investigations in the existing literature, this paper anchors the research topic in the interactive relationship between NEU and urban GI and systematically empirically analyzes its internal mechanism. Through the pioneering identification and quantification of the dual path of "quantity increase and quality improvement" of urban GI by NEU, this study has realized the expansion of the theoretical framework from a single dimension to a dual dimension, thus deepening the research on the economic effect of NEU and significantly enriching the academic achievements in the field of urban GI. (2) The existing research on NEU and GI mainly focuses on verifying a linear relationship or a single moderating effect. There are few studies on the following mechanisms underlying the multidimensional moderating effect and the nonlinear relationship between NEU and GI. This study not only verifies the positive effect of NEU on GI but also reveals its nonlinear characteristics and multivariate co-regulation mechanism (digital inclusive finance, government technology

support, informatization, regional economy, energy consumption). This is not only a deepening of Porter's hypothesis and endogenous growth theory in the context of green transformation, but also provides a new theoretical explanation dimension for understanding how institutional, financial and technological environments jointly shape the impact of urbanization on innovation and expands the content of urban development theory and GI theory. (3) Unlike the existing literature, which usually examines only a single moderating variable or the full-sample moderating effect, this paper designs a multi-dimensional, grouped-moderating-effect analysis framework that enables the simultaneous identification of multiple groups and multiple sources of heterogeneity, after controlling for city and year effects. The multivariate moderating effect helps reveal the internal logic and complex mechanisms by which NEU affects GI more comprehensively and deeply, and provides new ideas for future research. (4) Much of the international research is based in developed countries and focuses on urbanization and innovation in the post-industrial era. This paper, however, combines an analysis of China's NEU strategy—proposed in the new era—with the specific historical context of "economic transformation and development." By introducing government green support as an analytical perspective, this study reveals that NEU's path in urban GI exhibits significant nonlinearity. This conclusion not only confirms that the relationship between the two is far from summarised by a single linear trend but also provides key empirical evidence for understanding its inherent multiple equilibria and conditional dependence, thereby deepening relevant theoretical research. This provides important empirical evidence and new analytical perspectives for the development of NEU and GI theories.

## 2. Theoretical analysis and research hypotheses

The core meaning of urbanization goes beyond the simple regional transformation of population; it is a complex evolutionary process accompanied by industrial upgrading and social civilization. Therefore, its development path must place ecological civilization at its core and integrate green concepts throughout the process to achieve an intensive, intelligent, low-carbon, high-quality transformation. This shows that the NEU is a profound change guided by the new development concept, aiming to reshape the paradigm and logic of urban development. NEU goes beyond population concentration and spatial expansion, embodying a modern urban development paradigm characterized by economic growth, social harmony, and environmental sustainability. NEU represents a systematic reconfiguration of traditional urbanization, typically defined by "population mobility and urban expansion." GI, on the other hand, emphasizes the development and upgrading of green technologies aimed at resource recycling and environmental improvement; it encompasses creative activities across multiple dimensions, including technology, management, and institutional frameworks. From a conceptual perspective, NEU constitutes the practical context and demand engine for GI, while also providing a testing ground and incentive framework for GI, serving as a key driving force behind it.

Firstly, from the perspective of resource agglomeration theory, the spatial concentration of economic factors gives rise to a regional economic structure characterized by a clear division of labor and complementary strengths, with frequent business interactions, enabling more effective innovation. Compared with the traditional model, the NEU puts the "all-round development of people" in the first place [15], by improving the education level and literacy skills of the transferred population [16], it not only completed the transformation from "demographic dividend " to "talent dividend," but also directly reserved key human resources for scientific and technological innovation in new energy and intelligent manufacturing, and laid the quantitative foundation for output. In addition, the strategy strengthens the city's ability to absorb and integrate advanced production factors, such as capital and technology, by guiding optimal allocation of industry and infrastructure [17] and by laying the material basis for innovation and development. The rigid constraints of greening and high-quality development promote the structural transfer of capital, such as venture capital, bank loans, and government subsidies, and ensure that it flows accurately to urban GI projects. This not only provides sufficient financial guarantees for enterprises' R&D activities but also directly motivates increased R&D investment and expanded innovation boundaries. Meanwhile, the consumption upgrading effect caused by NEU, namely the growth of residents' income and the strengthening of environmental protection preferences [18], has spurred strong market demand for green, energy-saving, and high-tech

 

products. This strong derived demand is inversely transmitted through the market mechanism, serving as a key driver of urban GI output. To meet the strong demand for green and high-tech products driven by NEU, enterprises must increase their allocation of urban GI resources to expand their green product lines. In addition, the expansion of infrastructure such as urban construction, transportation, logistics and energy supply generated by the strategy has generated large-scale demand for green and low-carbon technologies [19], this cross-industry related demand, through technology lock-in and industrial chain transmission mechanism, encourages enterprises to actively research special technologies such as green buildings, thus directly realizing the accumulation of urban GI achievements in quantity.

Secondly, as an accelerator of modern factor flows, urbanization reshapes the spatial distribution of resources and factors [20]. The NEU has broken the path dependence of "winning by quantity " in traditional factor transfer and set higher standards for the connotation and quality of resource and factor flows. This advanced requirement is embodied in a non-balanced absorption mechanism in the talent dimension: the urban system prefers to screen and attract high-skilled, high-quality, innovative talent to build an intellectual capital core that supports high-quality development [21]. Relying on the dynamic learning effect of " Learning by Doing, " high-quality human capital can realize the rise of knowledge and the iterative evolution of skills in the process of practice, promote the development of technology R&D [22], and realize the fundamental paradigm shift of the innovation model, that is, from low value-added imitation to high value original innovation, which constitutes the core path of improving the quality of urban GI. In addition, the industrial ecological reconstruction caused by the NEU, by strengthening cross-industry technology spillovers and knowledge integration [23], provides a diversified technical intersection for urban GI and effectively broadens the possibility of technological breakthroughs. The integration of multi-technology fields (such as information, energy, and environmental technologies) not only gave rise to composite technology solutions but also substantially enhanced the core technological content and practical application value of urban GI achievements through the complementarity and inter-embedding of technologies. In addition, the NEU policy establishes strict environmental protection and industry norms by constructing an institutional environment aligned with urban GI, and internalizes external policy pressure as the core driving force for enterprises to improve technology and innovation quality [24]. Furthermore, endogenous growth theory holds that knowledge and technological information are the core drivers of sustained economic growth [25]. NEU brings together resources and factors within a specific geographical area to form a high-density, multi-stakeholder innovation network [26]. Various actors interact frequently within innovation networks; the industry-academia-research collaboration system formed within cities facilitates the rapid dissemination of knowledge and information, accelerating the transmission and recreation of technical knowledge, and leading to knowledge spillover effects and continuous innovation [27]. Constrained by the need to use the environment and resources sustainably, companies are compelled to seek alternative technologies that meet environmental standards, thereby driving innovation and the development of green technologies.

*Therefore, hypothesis 1: NEU promotes the "quantity increase" and "quality improvement" of GI.*

Digital inclusive finance (DIF) is a form of finance that primarily relies on digital technology to provide convenient financial products and services to various segments of society [28]. NEU cannot function without strong financial support. DIF can provide convenient, low-cost financial services to various stakeholders in NEU, particularly rural migrant workers, enterprises, and service providers [29]. As an important starting point for the structural reform of the financial supply side, DIF breaks through the service boundary of traditional finance through technological empowerment. It not only realizes the inclusive redistribution of financial resources between regions and entities, but also optimizes the combination of capital elements at a deeper level. It has laid a solid financial foundation for the quality and efficiency changes of the economic system. According to digital finance theory, financing constraints limit resource allocation and corporate innovation activities [30]. Supported by cutting-edge technologies such as big data, blockchain and AI, DIF effectively weakens and gradually eliminates the path dependence on traditional collateral and significantly improves the availability and convenience of financing for key subjects of urbanization, such as small and medium-sized enterprises and individual businesses [31]. However, in reality, there are significant regional

differences in DIF development, and construction levels in economically developed areas are generally high. This is mainly because its development is subject to the dual constraints and drive of regional economic foundation and information technology conditions, that is, the more developed the economy and the more advanced the technology, the stronger the penetration and coverage of DIF [32]. Through the DIF platform, R&D enterprises can easily secure financing to support GI activities—such as green R&D and the procurement of green equipment—thereby driving the production of green outcomes [33]. The literature shows that the deepening of DIF can not only improve the credit accessibility of the immigrant population in the dimensions of renting [34], education [35] and living consumption, but also expand the sources of funds for workers in employment and entrepreneurship [36], thus significantly promoting their social and economic integration and urban settlement willingness. At the same time, it is also used as a structural allocation mechanism to channel funds to green industries, intelligent manufacturing and eco-environmental protection projects, and it plays an indispensable capital-driven role in promoting the greening and upgrading of the industrial structure [37]. Moreover, by constructing an efficient information sharing and signal transmission platform, DIF not only significantly improves the docking efficiency of green technology achievements and terminal market demand, but also triggers a linkage effect at the industrial chain level and promotes upstream and downstream enterprises to carry out collaborative innovation in R&D, production and promotion [33]. On the contrary, the lag of regional DIF development will slow down the process of NEU construction, industrial upgrading and population flow to a certain extent [38], in particular, green industry and eco-city projects are difficult to attract social capital due to the lack of effective digital financing tools; clean energy, ecological restoration and other fields are also unable to obtain stable financial support due to the absence of targeted financing.

*Therefore, hypothesis 2: DIF has a complementary moderating effect, strengthening the impact of NEU on both "quantity increase" and "quality improvement" of urban GI.*

When exploring the endogenous drivers of NEU and urban GI, in addition to financial factors, the role of government technology support cannot be overlooked. The government is not only an institutional supplier, but also a resource allocator. By building a sound legal and policy system, and flexibly adopting tools such as financial subsidies, fund guidance, tax relief and risk sharing, it forms a multi-dimensional incentive for urbanization construction and urban GI activities [39], thus effectively promoting the quantity increase and quality improvement of innovation output. Environmental regulation theory posits that government intervention in pollution control promotes urban GI in the long run. Government technical support directly reduces the fixed costs and uncertainty associated with green R&D, increases firms' willingness to invest in R&D in sectors such as new energy, energy conservation, and environmental protection, and ensures that knowledge spillovers generated during the NEU process are more concentrated in environmentally friendly technologies [40]. Under the trend of increasingly complex innovation activities and deepening interdisciplinary integration, government science and technology support has not only become an important institutional arrangement to maintain the dominant position of enterprise innovation, but also a key path to improve the level of refinement and intelligence of urban governance [41]. In the face of rapid changes in the global economy, the process of informatization is deeply reshaping the logic of social operation and becoming the driving force behind the overall progress of society. Informatization is the process of applying modern information technology to optimize resource allocation, improve operational efficiency, and drive economic restructuring [42]. Informatization can build online platforms and networks to overcome geographical restrictions and has a positive effect on the equalization and popularization of public services [43]. Migrant workers can access convenient, efficient public services, which will help enhance smart urban management and support NEU's development. In the field of green energy material development, information technology enables technology embedding through algorithm optimization and virtual simulation, which not only shortens the time window from laboratory to industrialization but also reduces R&D costs [44] and accelerates the iteration speed of green technology from innovation to application. At the enterprise level, the penetration of information technology promotes the transformation of production systems toward

automation and intelligence. It improves resource allocation efficiency and R&D output by reducing human error and process waste [45]. At the same time, regional economic development has a significant structural impact on the spatial layout of NEU and the intensity of urban GI. Capital, technology, and talent are vital resources for economic development. Developed regions with abundant financial resources can drive high-intensity investment in green technology research and development and infrastructure construction. Its capital flow not only supports the clean energy and environmental protection industry but also enhances regional GI performance by improving urban public facilities and optimizing public services, and by building an institutional and material dual environment that supports the output of innovation results [46]. Developed cities realize orderly inflows and retention of high-level talent by optimizing the innovation ecosystem and enhancing the attractiveness of talent, thereby providing key knowledge supply and R&D kinetic energy for urban GI. In addition, due to differences in resource endowment, industrial structure and technological level, cities exhibit significant differences in energy consumption patterns. Urbanization requires large quantities of energy-intensive products such as steel and cement. As rural migrants move into urban areas, their energy consumption primarily consists of commercial energy sources such as electricity and natural gas, leading to an increase in energy consumption [47]. According to Porter's hypothesis, strict environmental regulation can stimulate the innovation compensation effect. Due to the intense pressure to reduce emissions and consumption in high-energy-consuming cities [48], the NEU process is often accompanied by more urgent demands for green transformation. To address the dilemma between environmental constraints and economic development, local governments tend to increase support for green technology research and development, thereby significantly increasing the marginal revenue and demand intensity of GI. NEU abandons the traditional extensive expansion mode and internalizes the concepts of sustainable development and green as core orientations, thereby driving the fundamental transformation of the urban energy structure from high-carbon dependence to low-carbon, high-efficiency systems [49]. In the process of NEU, high-energy-consuming cities will pay more attention to the development and utilization of ecological energy, which will help reduce the dependence on traditional energy, reduce carbon emissions [50], and promote the innovation and development of green energy technology.

*Therefore, hypothesis 3: DIF exhibits complementary moderating effects. The positive moderating role of DIF on the impact of NEU on urban GI is significantly strengthened by higher levels of government technology support, informatization, regional economic development, and energy consumption.*

Based on the above analysis, this paper identifies the primary pathways through which NEU influences GI, as well as the effects of moderating variables—including DIF, government technology support (GTS), informatization (INT), regional economic development (AGDP), and energy consumption (ENC)—on NEU and GI. The relationships among these variables are illustrated in Fig 1 (where DIF × GTS, DIF × INT, DIF×AGDP, and DIF × ENC denote interaction-moderating effects).

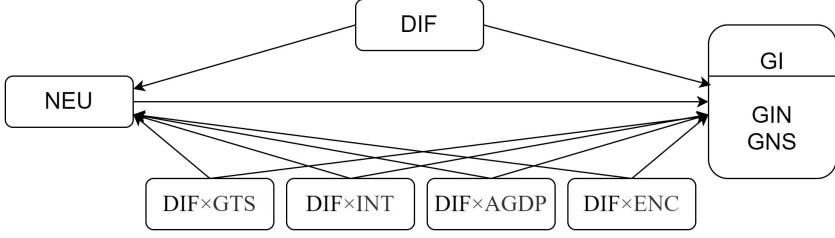

**Fig 1. The influence path diagram of variables.**

# 3. Research design

## 3.1. Indicator design

### i) Dependent variable

The dependent variable is urban GI. To reflect the "quantity increase" of urban GI (GIN) and "quality improvement" of urban GI (GNS), this study uses the total number of green patent applications per 10,000 people in the cities to measure GIN and the number of green invention patent applications per 10,000 people in the cities to measure GNS [51]. The total number of green patent applications not only reflects the state of innovation activities but is also closely related to the economic and environmental benefits of GI. Green patent data provides a specific, tangible measurement standard for innovation activities. The number of applications reflects the innovation efforts and intentions. The number of green patent applications can not only better measure the process of GI, but also directly reflect the research and development results of enterprises, scientific research institutions or individuals in the field of green technology. The number of green patent applications is standardized, quantifiable and internationally comparable, which is convenient for statistical analysis and cross-country comparisons. This choice is consistent with the mainstream literature on environmental innovation and ecological innovation [51,52]. To mitigate the effect of a few extreme values, GIN and GNS data are winsorized at the 1% level.

### ii) Independent variable

This paper selects NEU as the core explanatory variable. In terms of its theoretical connotation and structural characteristics, this variable covers four main aspects: population, economy, environment and society [5]. The measurement indicators are listed in Table 1, in which the Engel coefficient ratio, the urban-rural income ratio and the urban-rural consumption

Table 1. Indicators of NEU.

| First-level Indicators | Second-level Indicators | Code | Third-level Indicators |
|---|---|---|---|
| NEU | Population urbanization | X1 | Urbanization rate of permanent population |
| | | X2 | Number of participants in urban basic medical insurance |
| | | X3 | Urbanization rate of registered residence |
| | | X4 | Proportion of general public budget expenditure on social security and employment |
| | | X5 | Urban pension insurance coverage rate |
| | Economic Urbanization | X6 | Population density in urban areas |
| | | X7 | The ratio of Engel coefficient between urban and rural areas |
| | | X8 | The logarithm of economic agglomeration degree |
| | | X9 | Urban rural consumption ratio |
| | | X10 | Per-capita Disposable Income of Urban Residents |
| | | X11 | Urban rural income ratio |
| | Environmental Urbanization | X12 | Urban per capita road area |
| | | X13 | Harmless treatment rate of urban household waste |
| | | X14 | Urban per capita construction land area |
| | | X15 | Green coverage rate of built-up areas |
| | | X16 | Per capita built-up area of urban areas |
| | | X17 | Urban sewage treatment rate |
| | Social urbanization | X18 | Proportion of hospital hygiene beds to urban permanent population |
| | | X19 | Proportion of regular undergraduate and junior college students to registered population |
| | | X20 | Urban water supply penetration rate |
| | | X21 | The total collection of books in public libraries |
| | | X22 | Urban gas penetration rate |

ratio are all negative indicators. To eliminate the directional difference, the research has conducted a reciprocal conversion. Then, using the entropy-weight method, the weights and information concentration of the multi-index system are calculated, thereby constructing the NEU comprehensive score for regression analysis.

The specific calculation process of the entropy weighting method is as follows.

The original data matrix is:

$$X = (x_{ij})_{m \times n}, \quad i = 1, 2, \ldots, m; \ j = 1, 2, \ldots, n \tag{1}$$

n is the cities, m is the number of indicators. $x_{ij}$ is the value of the jth indicator for the ith sample.

Normalizing data:

$$p_{ij} = \frac{x_{ij} - \min(x_j)}{\max(x_j) - \min(x_j)} \tag{2}$$

Calculating weights:

$$q_{ij} = \frac{p_{ij}}{\sum_{i=1}^{m} p_{ij}} \tag{3}$$

Calculating information entropy:

$$e_j = -k \sum_{i=1}^{m} q_{ij} \ln(q_{ij}), \quad k = \frac{1}{\ln m} \tag{4}$$

Calculating the coefficient of variation:

$$d_j = 1 - e_j \tag{5}$$

Calculating the entropy ratio:

$$w_j = \frac{d_j}{\sum_{j=1}^{n} d_j} \tag{6}$$

Calculating the composite score:

$$S_i = \sum_{j=1}^{n} w_j \cdot p_{ij} \tag{7}$$

To verify the reliability and correlation of the third-level indicators of NEU, the data reliability is first checked. The Cronbach's alpha coefficient measures the reliability of the data, and the coefficient is calculated for each of the three-level metrics and for all the metrics. The results are shown in Table 2. The Cronbach's alpha values for each metric are greater than 0.7, and the overall reliability of the data is also greater than 0.7, indicating that the data meet the reliability requirements. Secondly, KMO and Bartlett's test of sphericity are used to assess the partial correlations and the correlation matrix among the variables, as shown in Table 3. The result of the test is that the overall value of KMO is 0.7792 > 0.6, Bartlett's test of sphericity passes the test at the 0.01 level, indicating a good correlation and validity between the variables.

### iii) Control variables (CV)

Control variables include foreign direct investment (FDI), openness to the outside (OOW), human capital level (HCL), Internet penetration rate (INT), per capita volume of road freight transportation (RFV), and level of social consumption

**Table 2. Cronbach's Alpha of Indicators of NEU.**

| Variable | Cronbach's alpha | Variable | Cronbach's alpha | Variable | Cronbach's alpha |
|----------|------------------|----------|------------------|----------|-------------------|
| X1 | 0.816 | X12 | 0.8293 | NEU | 0.8380 > 0.7 |
| X2 | 0.8266 | X13 | 0.8339 | | |
| X3 | 0.8332 | X14 | 0.8302 | | |
| X4 | 0.8428 | X15 | 0.8319 | | |
| X5 | 0.828 | X16 | 0.8181 | | |
| X6 | 0.8253 | X17 | 0.8314 | | |
| X7 | 0.8377 | X18 | 0.8357 | | |
| X8 | 0.8395 | X19 | 0.8312 | | |
| X9 | 0.819 | X20 | 0.8266 | | |
| X10 | 0.8334 | X21 | 0.8406 | | |
| X11 | 0.8319 | X22 | 0.8467 | | |

**Table 3. KMO and Bartlett Test of Sphericity Results.**

| Variable | KMO | Variable | KMO | Variable | KMO | Bartlett test of sphericity | |
|----------|-----|----------|-----|----------|-----|------------------------------|---|
| X1 | 0.7854 | X12 | 0.8509 | Overall | 0.7792 > 0.6 | Chi-square | 33184.396*** |
| X2 | 0.7905 | X13 | 0.8999 | | | | |
| X3 | 0.5740 | X14 | 0.8951 | | | | |
| X4 | 0.6430 | X15 | 0.8423 | | | | |
| X5 | 0.8821 | X16 | 0.8408 | | | | |
| X6 | 0.8571 | X17 | 0.7079 | | | | |
| X7 | 0.4807 | X18 | 0.8400 | | | | |
| X8 | 0.6819 | X19 | 0.7233 | | | | |
| X9 | 0.8891 | X20 | 0.7768 | | | | |
| X10 | 0.6446 | X21 | 0.6939 | | | | |
| X11 | 0.8845 | X22 | 0.6384 | | | | |

(SCL). FDI is measured by the proportion of foreign direct investment to GDP, TEL is measured by science and technology expenditures, OOW is measured by the proportion of total import and export volume to GDP, HCL is measured by the proportion of regular undergraduate and junior college students in the total population at the end of the year, RFV is measured by the ratio of road freight transportation volume to total resident population, the ratio of total retail sales of consumer goods to regional gross domestic product measures SCL.

### iv) Moderating variables

To reveal the changing rules governing the relationships between explanatory and explained variables across different situations, this study constructs a multivariate moderating effect model that includes main and group moderating variables. The main moderating variable is Digital Inclusive Finance (DIF), and the data are from the Digital Inclusive Finance Index released by Peking University. The grouping moderating variables cover government technology support (GTS), informatization (INT), regional economic development (AGDP), and energy consumption (ENC), aiming to test the heterogeneity of the adjustment effect across groups or interactions. GTS is measured by the ratio of science expenditures in the general public budget, AGDP is measured by per capita GDP, the Internet penetration rate measures INT and the product of the ratio of the sum of primary and secondary production to GDP and coal consumption measures ENC.

## 3.2. Models design

To explore the impact of NEU on urban GI, the benchmark regression model is constructed as follows:

$$GIN_{it} = \alpha_0 + \beta_1 NEU_{it} + \gamma_1 CV_{it} + \delta_i + \theta_t + \varepsilon_{it} \tag{8}$$

$$GNS_{it} = \alpha_1 + \beta_2 NEU_{it} + \gamma_2 CV_{it} + \delta_i + \theta_t + \varepsilon_{it} \tag{9}$$

Considering that the independent variables may be endogenous, the two-stage instrumental variable method is employed to construct the regression model. NEU lagged one period (LAG_NEU), and nighttime lighting value (NLV) is an instrumental variable:

$$NEU_{it} = \alpha_2 + \beta_3 LAG\_NEU_{it} + \gamma_3 CV_{it} + \delta_i + \theta_t + \varepsilon_{it} \tag{10}$$

$$GIN_{it} = \alpha_3 + \beta_4 (NEU'_{it} + \sigma) + \gamma_4 CV_{it} + \delta_i + \theta_t + \varepsilon_{it} \tag{11}$$

$$GNS_{it} = \alpha_4 + \beta_5 (NEU'_{it} + \sigma) + \gamma_5 CV_{it} + \delta_i + \theta_t + \varepsilon_{it} \tag{12}$$

$$NEU_{it} = \alpha_5 + \beta_6 NLV_{it} + \gamma_6 CV_{it} + \delta_i + \theta_t + \varepsilon_{it} \tag{13}$$

$$GIN_{it} = \alpha_6 + \beta_7 (NEU'_{it} + \sigma) + \gamma_7 CV_{it} + \delta_i + \theta_t + \varepsilon_{it} \tag{14}$$

$$GNS_{it} = \alpha_7 + \beta_8 (NEU'_{it} + \sigma) + \gamma_8 CV_{it} + \delta_i + \theta_t + \varepsilon_{it} \tag{15}$$

To further investigate the mechanism and boundary conditions of NEU on urban GI, this paper introduces the interaction term between DIF and NEU ($cDIF_{it} \times cNEU_{it}$). It conducts group regression using GTS, INT, AGDP and ENC for sample cities to test the heterogeneity of the moderating effect. The following moderating effect model is constructed:

$$GIN_{it} = \alpha_8 + \beta_9 NEU_{it} + \rho_1 DIF_{it} + \varphi_1 cDIF_{it} \times cNEU_{it} + \gamma_9 CV_{it} + \delta_i + \theta_t + \varepsilon_{it} \tag{16}$$

$$GNS_{it} = \alpha_9 + \beta_{10} NEU_{it} + \rho_2 DIF_{it} + \varphi_2 cDIF_{it} \times cNEU_{it} + \gamma_{10} CV_{it} + \delta_i + \theta_t + \varepsilon_{it} \tag{17}$$

$\alpha$ is constant, $\beta$ is the coefficient of the independent variable, $\delta$ is city fixed effect, $\theta$ is year fixed effect, and $\varepsilon$ is random error.

## 3.3. Data sources and descriptive statistics

This paper uses panel data from 284 cities spanning 2011–2022. The data for GIN and GNS are from the Chinese Research Data Services Platform (CNRDS), NEU encompasses multiple second-level indicators, and the data for these indicators mainly originate from the China Urban Statistical Yearbook, statistical yearbooks of various regions, the EPS database and the China Energy Statistical Yearbook. The data for control variables FDI, TEL, OOW, HCL, RFV and SCL

are from the China Urban Statistical Yearbook, and the data for INT is from the EPS database. The DIF data are sourced from the Digital Financial Inclusion Index published by Peking University. The data for the moderating variables GTS, INT and AGDP are from the China Urban Statistical Yearbook, and the data for ENC are from the China Energy Statistical Yearbook. The data for GES is from the zldatas database. A small number of missing values in NEU, FDI and ENC are complemented by linear interpolation. To prevent extreme values from affecting regression accuracy, the GIN and GNS data have been winsorized at the 1% level to eliminate potential abnormal fluctuations. Table 4 shows the results of descriptive statistical analysis of the sample data. To enable further empirical analysis, the main independent variable, NEU and the dependent variables, GIN and GNS, are standardized. The standardized coefficients can directly compare the influence degree of variables, so as to improve the interpretability of the model results.

To further test multicollinearity, this paper calculates the VIF (variance inflation factor) values for the independent variable NEU and the control variables FDI, INT, RFV, HCL, OOW and SCL. As shown in Table 5, the VIF values of all variables are less than 5, and the correlation coefficients between each variable are also less than 0.7. It is considered that there is no multicollinearity problem between NEU and the control variables.

## 4. Empirical analysis

### 4.1. Benchmark regression

Table 6 reports the benchmark regression results for NEU on "quantity increase and quality improvement" in urban GI. The coefficients of NEU to GIN and GNS are 0.8126 ($p < 0.01$) and 0.8315 ($p < 0.01$) without controlling for other variables (E1 and E2). After introducing control variables such as FDI, TEL, OOW, HCL, INT, RFV and SCL (E3 and E4), the coefficients of NEU to GIN and GNS are 0.7452 ($p < 0.01$) and 0.7892 ($p < 0.01$), respectively, which are significant at the 1% level, indicating that NEU has a significant positive effect on the "quantity increase and quality improvement" of urban GI. This means that for every 1 unit increase in NEU, GIN increases by an average of 0.7452 units, GNS increases by an average of 0.7892 units, the adj.R2 in E1 and E3 increases from 0.462 to 0.861, and the adj.R2 in E2 and E4 increases

**Table 4. Descriptive Statistical Results.**

| VAR | Mean | SD | Max | Min | N |
|---|---|---|---|---|---|
| NEU | 0.1209 | 0.0562 | 0.5371 | 0.0354 | 3408 |
| GIN | 0.9391 | 1.3566 | 7.7034 | 0.0168 | 3408 |
| GNS | 0.4085 | 0.6777 | 4.0229 | 0.0048 | 3408 |
| RFV | 32.2915 | 65.9670 | 3042.3968 | 1.2166 | 3408 |
| OOW | 0.1758 | 0.2811 | 2.4913 | −0.0201 | 3408 |
| HCL | 0.0200 | 0.0250 | 0.1502 | −0.0006 | 3408 |
| INT | 25.6078 | 14.9261 | 125.8681 | 0.3796 | 3408 |
| FDI | 0.0148 | 0.0185 | 0.2827 | −0.2186 | 3396 |
| SCL | 0.3817 | 0.1094 | 1.0126 | 0 | 3408 |
| NGIN | 0.6549 | 1.0162 | 9.9852 | 0 | 3408 |
| NGNS | 0.1115 | 0.2471 | 4.3563 | 0 | 3408 |
| CIT | 0.2312 | 0.0925 | 0.7392 | 0.0851 | 3408 |
| NLV | 8.8598 | 9.9050 | 60.0526 | 0.1297 | 3408 |
| DIF | 226.3505 | 83.2703 | 581.2300 | 2.7000 | 3408 |
| AGDP | 5.6743 | 3.3874 | 25.8482 | 0.2134 | 3408 |
| GTS | 0.0172 | 0.0173 | 0.1423 | 0.0001 | 3408 |
| GES | 0.0075 | 0.0039 | 0.0256 | 0.0004 | 3408 |
| ENC | 6.4742 | 0.7373 | 8.8560 | 4.0950 | 3408 |

**Table 5. The variance inflation factor and correlation coefficient of each variable.**

|  |  | NEU | FDI | INT | RFV | HCL | OOW | SCL |
|---|---|---|---|---|---|---|---|---|
| Variance Inflation Factor | VIF | 2.94 | 1.17 | 1.27 | 1.01 | 1.92 | 1.55 | 1.04 |
|  | 1/VIF | 0.3398 | 0.8536 | 0.7876 | 0.9881 | 0.5211 | 0.6452 | 0.9572 |
| correlation coefficient | NEU | 1 |  |  |  |  |  |  |
|  | FDI | 0.2341 | 1 |  |  |  |  |  |
|  | TEL | 0.5943 | 0.3153 |  |  |  |  |  |
|  | INT | 0.4203 | −0.0321 | 1 |  |  |  |  |
|  | RFV | 0.0176 | −0.0254 | 0.0225 | 1 |  |  |  |
|  | HCL | 0.6714 | 0.2274 | 0.3228 | 0.0117 | 1 |  |  |
|  | OOW | 0.5489 | 0.2205 | 0.2021 | −0.0250 | 0.2783 | 1 |  |
|  | SCL | 0.0850 | −0.0203 | 0.0772 | −0.0814 | 0.1467 | −0.0531 | 1 |

**Table 6. Benchmark Regression Results.**

| VAR | E1 | E2 | E3 | E4 |
|---|---|---|---|---|
|  | GIN | GNS | GIN | GNS |
| NEU | 0.8126*** (6.4517) | 0.8315*** (5.2419) | 0.7452*** (8.7361) | 0.7892*** (9.0166) |
| CV | N | N | Y | Y |
| cons | −0.1928*** (−4.2662) | −0.1272** (−2.2558) | 0.5144*** (7.8443) | 0.5625*** (8.0641) |
| N | 3408 | 3408 | 3396 | 3396 |
| adj. $R^2$ | 0.462 | 0.344 | 0.861 | 0.849 |
| id FE | Y | Y | Y | Y |
| Year FE | Y | Y | Y | Y |

Note: t statistics in parentheses, * $p < 0.1$, ** $p < 0.05$, *** $p < 0.01$

from 0.344 to 0.849. The overall fitting degree of the model is improved, and the control variables such as FDI, INT, RFV, HCL, OOW and SCL in the model explain the 'residual variation', which leads to the improvement of the overall fitting degree of the model. This further verifies that NEU is an important driving force for GIN and GNS and shows that the model effectively captures a broader set of driving factors. The evidence suggests that NEU catalyzes urban GI, positively impacting both "quantity increase" and "quality improvement.". The comparison of E3 and E4 shows that the coefficients of NEU are 0.7452 and 0.7892, the "quality improvement" effect of NEU on GI is stronger than the "quantity increase" effect of NEU on GI, and NEU in China is in a transitional phase from "quantity-based growth" to "quality-oriented enhancement". This shows that China is shifting from the traditional ' scale-driven ' growth to the ' innovation-driven ' development in the new era. Urbanization is no longer a simple population migration. NEU is actively exerting the effects of factor agglomeration and high-quality innovation, and guiding the industry toward green, low-carbon, and modernization. This also supports the viewpoints of endogenous growth theory and innovation theory, and the spatial agglomeration of economic activities effectively promotes innovation [53]. The industrial upgrading and environmental regulation brought by NEU have forced enterprises to actively seek fundamental changes and GI to gain a competitive advantage. As a key link, the "quality improvement" of GI can better reflect the deep value of NEU than the "quantity increase." NEU has created market opportunities and factor endowment conditions for the increment and improvement of urban GI [54]. Moreover, the strict ecological environment standards and efficient development orientation under the NEU paradigm have formed a severe external compliance pressure on enterprises, forcing enterprises to improve total factor productivity through

technological innovation and process reengineering, thus promoting GI [55]. This conclusion systematically deconstructs the quantity increase and quality improvement of urban GI within the same framework for the first time, thereby expanding the theoretical boundary of the urbanization-innovation relationship and enabling a more accurate explanation of the practical needs of 'quantity and quality' in green transformation. *Hypothesis 1* is verified.

To further verify that the explanatory variable NEU affects the GIN and GNS models and that the models meet the requirements of normality, chi-square, and independence, this paper evaluates the plots of the fitted distributions of the residuals from regressions of NEU on GIN and GNS. First, the fitted values and residuals are calculated for the regression results of NEU affecting GIN, and the plots of residual fitted values are shown in Fig 2. The residual points are randomly distributed around the 'zero residual line' without an obvious trend, indicating that the 'linear hypothesis' of the model is supported. When the fitting value changes from −4 to 4, the dispersion of the residuals does not systematically expand or contract, indicating that the model's homoscedasticity hypothesis holds. Plotting the normal probability of the residuals (Q-Q plots) meets the linear regression model's requirement for normality. Next, the fitted values and residuals are calculated for the regression results of NEU affecting GNS, and the residual fitted values are plotted, as shown in Fig 3. The residual points in the figure are scattered around the zero residual line, and there is no systematic bending or monotonic trend with the fitting value. The model's linear hypothesis is basically established. It is observed that there is no trend of "trumpet shape" or "inverted trumpet shape" in the range of fitting values from −4 to 4, and the homoscedasticity hypothesis of the model is basically established. Plotting the normal probability of the residuals (Q-Q plot) meets the requirement of residual normality for the linear regression model.

## 4.2. Robustness test

### i) Removing special city samples

As important nodes of urbanization and innovation, China's four municipalities (Beijing, Chongqing, Shanghai, Tianjin) are in the leading position in terms of population agglomeration and innovation output. Due to its special position in the sample, it may have a non-random impact on the estimation of NEU innovation effects. Therefore, this paper excludes the

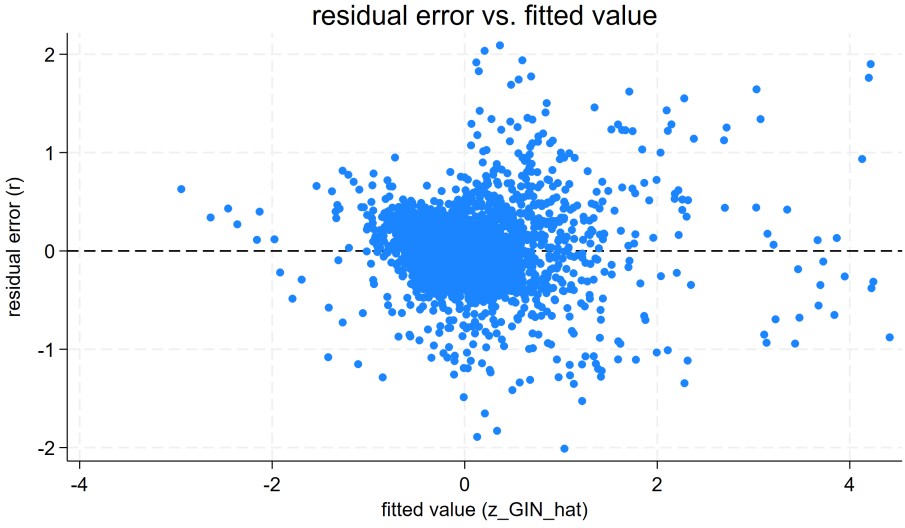

**Fig 2. NEU Affects GIN's Residual Error vs. Fitted Value Graph.**

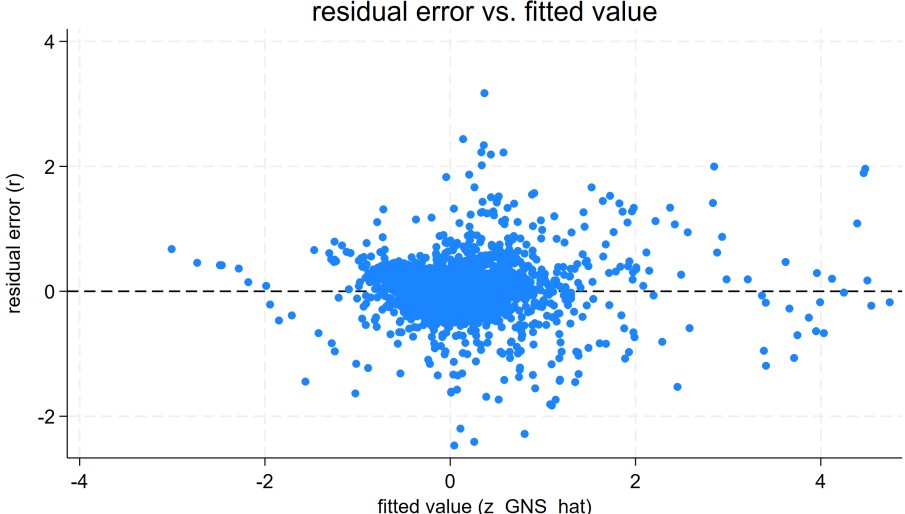

**Fig 3. NEU Affects GNS's Residual Error vs. Fitted Value Graph.**

samples of these four municipalities in the robustness test to examine the consistency of the estimation results. The relevant regression results are shown in Table 7. After controlling for other variables and fixed effects, the NEU coefficients for models E5 and E6 are 0.7260 and 0.7394, respectively, and both are significant at the 1% level. The results confirm that even when municipalities with significant innovation agglomeration characteristics and population absorption capacity are excluded, the promotion effect of NEU on urban GI "quantity increase" and "quality improvement" remains significant, and the benchmark conclusion is robust and reliable.

### ii) Removing special year samples

Considering that the NEU strategy was proposed in 2015 and promoted nationwide in 2016, the institutional adaptation and data characteristics in the early stage of the policy may differ systematically from those in the subsequent stage, thereby affecting the consistency of the estimation results. Therefore, this paper excludes the 2015 and 2016 samples from the robustness test to control for the interference of unconventional factors during the policy start-up period. The regression results for E7 and E8 in Table 7 showed that the coefficients for NEU to GIN and GNS are 0.8031 and 0.8433, respectively, and both are significant. The results confirm that even after excluding the initial sample of the policy, the promotion effect of NEU on GIN and GNS remains significant, and the benchmark conclusion is robust and reliable.

### iii) Adding dummy variables

While examining the impact of NEU on urban GI, other policy and structural variables may also affect it. In this paper, the new energy city policy (NEM) is further introduced into the model as a control variable to control its direct impact on urban GI, so as to more accurately identify the net effect of NEU. The core purpose of NEM is to promote the green and low-carbon transformation of urban energy structure and explore a new model of sustainable urban energy development, which has a positive effect on GI [56]. Including this policy in the regression model partially absorbs policy-related regional characteristics. It can control for exogenous effects, such as resource tilt and institutional support, that the policy may introduce, thereby testing whether the regression results are sensitive to these characteristics. In this paper, NEM is set as a dummy variable. The city that implements the policy is assigned 1, and the city that does not implement the policy is assigned 0. The variable is included in the regression model. The relevant estimation results are shown in Table 7. After controlling NEM, the estimation results for models E9 and E10 show that the regression coefficients for NEU on urban GIN and GNS

**Table 7. Results of Robustness Test I.**

| VAR | E5 | E6 | E7 | E8 | E9 | E10 |
|---|---|---|---|---|---|---|
| | GIN | GNS | GIN | GNS | GIN | GNS |
| NEU | 0.7260*** (10.7600) | 0.7394*** (10.5814) | 0.8031*** (8.4162) | 0.8433*** (8.5949) | 0.7349*** (8.6252) | 0.7778*** (8.8980) |
| CV | Y | Y | Y | Y | Y | Y |
| NEM | | | | | 0.1507*** (3.5072) | 0.1656*** (3.5615) |
| cons | 0.4666*** (6.9475) | 0.5091*** (7.1457) | 0.5228*** (7.1979) | 0.5528*** (7.2594) | 0.4837*** (7.2731) | 0.5289*** (7.3464) |
| N | 3348 | 3348 | 2830 | 2830 | 3396 | 3396 |
| adj. $R^2$ | 0.855 | 0.840 | 0.852 | 0.842 | 0.862 | 0.850 |
| id FE | Y | Y | Y | Y | Y | Y |
| Year FE | Y | Y | Y | Y | Y | Y |

Note: See notes for Table 6.

are 0.7349 and 0.7778, respectively, and both are significant. This shows that, after accounting for the impact of NEM, the "quantity increase" and "quality improvement" effects of NEU on urban GI remain significant and robust.

**iv) Replacing independent variable**

Since the core independent variable NEU is synthesized from multiple indicators, if the indicator screening is unreasonable or the measurement method has a systematic error, this will lead to measurement errors, which in turn threaten the unbiasedness and consistency of the regression coefficients. In this paper, the entropy method is used to calculate the comprehensive score of NEU for regression analysis, but any measurement method has its limitations. It is well known that there are differences between the entropy method and the factor analysis method in application scope and characteristics [57]. This paper needs to test whether the conclusion is affected by the specific weighting method. If the impact results are highly consistent in both statistical significance and economic significance under the two measurement methods, it can be effectively proved that the impact effect comes from the data itself, rather than the results determined by a specific algorithm [58]. This paper uses exploratory factor analysis to remeasure NEU as an explanatory variable (NEU_EFA) for inclusion in the regression model, and the results are shown in E11 and E12 of Table 8. The coefficient of NEU_EFA on GIN is 0.4543, and the coefficient of NEU_EFA on GNS is 0.5784, both of which pass the test of significance, verifying the robustness of the regression results. This confirms that this effect is a common feature of the NEU process, rather than an accidental product of a single indicator.

**v) Replacing dependent variable**

Measurement bias in dependent variables is a potential source of regression bias. Although this paper uses the number of green patent applications as a proxy for urban GI, in reality, non-substantive innovation behaviors may be carried out to secure preferential policies, resulting in noise in the indicator's capture of the true level of innovation. In order to alleviate this problem, this paper adopts a new measurement scheme: the number of urban GI is measured by the total number of green patents granted per 10,000 people (NGIN), and the quality of urban GI is measured by the total number of green invention patents granted per 10,000 people (NGNS). The estimation results in Table 8 show that the coefficients for NEU to NGIN and NGNS in E13 and E14 are 0.7452 and 0.2602, respectively, and both are significant, indicating that the "quantity increase" and "quality improvement" effects of NEU on urban GI remain valid after index replacement. In addition, the NEU coefficient remains positive and significant across different measurement methods, further indicating that the research conclusion is robust to the choice of urban GI proxy variables and is not driven by specific indicators.

**Table 8. Results of Robustness Test II.**

| VAR | E11 | E12 | E13 | E14 | E15 | E16 |
|---|---|---|---|---|---|---|
|  | GIN | GNS | NGIN | NGNS | GIN | GNS |
| L2. GIN |  |  |  |  | 0.4723***<br>(30.7362) |  |
| L2. GNS |  |  |  |  |  | 0.4318***<br>(28.0211) |
| NEU |  |  | 0.7452***<br>(8.7361) | 0.2602***<br>(3.9426) | 0.0792*<br>(1.7857) | 0.1486***<br>(3.2061) |
| CIT |  |  |  |  |  |  |
| NEU_EFA | 0.4543***<br>(6.4812) | 0.5784***<br>(7.6748) |  |  |  |  |
| CV | Y | Y | Y | Y | Y | Y |
| cons | 0.4383***<br>(6.5616) | 0.4877***<br>(6.8886) | 0.5144***<br>(7.8443) | 0.2951***<br>(7.9299) | 0.3413***<br>(6.6025) | 0.3692***<br>(6.7889) |
| N | 3396 | 3396 | 3396 | 3396 | 2830 | 2830 |
| adj. $R^2$ | 0.852 | 0.840 | 0.861 | 0.765 | 0.921 | 0.914 |
| id FE | Y | Y | Y | Y | Y | Y |
| Year FE | Y | Y | Y | Y | Y | Y |

Note: See notes for Table 6

## vi) Dynamic two-way fixed effects analysis

To prevent the serious over-identification problem caused by too many lag orders and excessive expansion of instrumental variables in GMM analysis, and to alleviate the long-term persistence and path-dependence problems that may arise in dependent variables, this paper draws on the existing literature. Based on the benchmark two-way fixed-effect model, a second-order lag term for the dependent variable (L2.GIN and L2.GNS) is introduced to construct a dynamic panel model for robustness testing [59,60]. Moreover, the second-order lag can capture the long-term dynamic characteristics more fully than the first-order lag. The results (in Table 8) show that the coefficients of the second-order lag term L2.GIN and L2.GNS are 0.4723 and 0.4318, respectively, which are significantly positive, indicating that GIN and GNS exhibit significant long-term path dependence, and that the historical state has a continuous and stable positive impact on the current GI. After controlling for long-term dynamic inertia, city fixed effects, and year fixed effects, the coefficients for NEU on GIN and GNS are 0.0792 and 0.1486, respectively, and are significant, with coefficients consistent with the benchmark regression. It shows that the dynamic double fixed-effect model not only does not negate the benchmark regression's conclusion but also further supports the robustness of the results. Therefore, this paper still uses the static two-way fixed effect model as the core conclusion, and the dynamic model is only a robustness supplement.

## 4.3. Endogenous analysis

The previous empirical results show that NEU can effectively promote the "quantity increase" and "quality improvement" of urban GI. However, in reality, cities with higher GI levels often perform well in NEU development [61], which means that there may be a two-way causal relationship between the two, which leads to potential endogenous bias problems. Therefore, by employing an instrumental variable approach to address endogeneity, we identify a causal effect of NEU, leading to a significant positive effect on GIN and GNS. Moreover, if the important covariates closely related to the core explanatory variable (NEU) and the explanatory variable (GI) are not included in the model, this may lead to bias due to missing

variables, affecting the unbiasedness and consistency of the estimation results. Accordingly, we adopt the instrumental variable method to mitigate potential endogenous biases and ensure the credibility of the results.

### i) The independent variable lagged one period (LAG_NEU) is used as an instrumental variable

Since time is irreversible, the predetermined variable is not affected by the current period [62]. LAG_NEU is a predetermined variable; NEU in the t-1 period cannot be affected by the NEU in the t period. There is no reverse causality between GI and LAG_NEU. LAG_NEU is not related to the current error term. LAG_NEU satisfies the core hypothesis of instrumental variable exogeneity. At the same time, city fixed effects and year fixed effects are introduced into the model to control for the possibility that urban characteristics may be related to both LAG_NEU and GI. While LAG_NEU may be correlated with past realizations of the endogenous variable, it satisfies the exclusion restriction because it influences the current outcome solely through its contemporaneous effect on NEU, the endogeneity of the regression is verified via the instrument LAG_NEU. Table 9 shows the 2SLS regression results. The first stage (E17) shows that the coefficient on LAG_NEU is 0.4786 and is statistically significant, verifying the correlation between the instrumental variables and the endogenous variables. In the second stage (E18 and E19), the coefficients of the NEU estimator on GIN and GNS are 1.3001 and 1.3665, respectively, and both are significant, indicating that after correcting endogenous bias, NEU still has a significant role in promoting the "quantity increase" and "quality improvement" of urban GI, and the benchmark conclusion remains robust, LM statistic is 639.126, $p < 0.01$, underidentification test passes, Wald $F = 820.614 > 15$, and weak instrumental variable test passes. After controlling for potential endogeneity, the empirical results still show that NEU plays a significant role, and the benchmark conclusion remains robust.

### ii) Night lighting value (NLV) as an instrumental variable

Nighttime light value (NLV) can objectively reflect the spatial distribution and intensity of human activities within the city, and its brightness level is positively correlated with the degree of population agglomeration [63], as an important driving force and manifestation of the urbanization process, population agglomeration makes the NLV intrinsically linked to the NEU level in theory, so as to meet the correlation requirements between instrumental variables and endogenous explanatory variables. NLV has natural properties and primarily captures light emitted by infrastructure, industrial production and

**Table 9. Results of endogeneity test.**

| VAR | E17 | E18 | E19 | E20 | E21 | E22 |
|---|---|---|---|---|---|---|
| | NEU | GIN | GNS | NEU | GIN | GNS |
| NEU | | 1.3001*** (12.7022) | 1.3665*** (12.8566) | | 9.7819*** (6.2855) | 9.1890*** (6.2935) |
| LAG_NEU | 0.4786*** (28.6513) | | | | | |
| NLV | | | | 0.0087*** (6.0542) | | |
| CV | Y | Y | Y | Y | Y | Y |
| N | 3113 | 3113 | 3113 | 3396 | 3396 | 3396 |
| adj. $R^2$ | | 0.382 | 0.230 | | | |
| id FE | Y | Y | Y | Y | Y | Y |
| Year FE | Y | Y | Y | Y | Y | Y |
| Anderson canon. corr. LM statistic | 639.126*** | | 639.126*** | 36.334*** | | 36.334*** |
| Cragg-Donald Wald F statistic | 820.614 > 15 | | 820.614 > 15 | 36.551 > 15 | | 36.551 > 15 |

Note: See notes for Table 6

commercial activities. It is generally directly observed by satellite remote sensing. NLV reflects the objective geographical distribution of energy consumption and human activities, and is not affected by local government policy intervention or economic activities [64]. Moreover, the causal relationship between NLV's impact on the environment and GI activities, such as technological progress and R&D investment, is extremely weak and indirect, and does not directly affect GI, which is an exogenous variable. Most of the existing researchers use NLV as a proxy variable or instrumental variable of NEU for endogenous analysis [65,66]. Therefore, endogeneity is tested using NLV as an instrumental variable for NEU. E20 shows that the coefficient of NLV on NEU is 0.0087; NLV has a strong correlation with NEU. E21 shows the coefficient of NEU on GIN is 9.7819, and E22 shows the coefficient of NEU on GNS is 9.1890, both of which pass the test of significance. The fixed effect absorbs a large number of variations, resulting in a very low $R^2$, which is a common phenomenon in the high-dimensional fixed effect model and does not affect the consistency of the coefficients. LM statistic is 36.334, $p < 0.01$, underidentification test passes. Wald $F = 36.551 > 15$ weak instrumental variable test passes. After adjusting for endogenous tests, such as the instrumental variable method, the promotion effect of NEU on urban GI in the quantitative and quality dimensions remains significant, and the benchmark conclusion has not changed substantially.

### 4.4. Moderating effects

The "quantity increase" and "quality improvement" effects of NEU on urban GI have been verified in the baseline regression, but their effects are obviously differentiated among different cities. This differentiation is closely related not only to the city's own development stage and strategic positioning, but also to multiple factors such as DIF, GTS, INT, AGDP and ENC. In particular, digital inclusive finance, which has the characteristics of low thresholds, wide coverage and high efficiency compared with traditional finance, has injected new vitality into the urban economy and may also change NEU's path toward urban GI. Based on this, this paper selects DIF as the main moderating variable and uses GTS, INT, AGDP and ENC as grouping variables to systematically investigate the moderating effect of NEU on urban GI.

#### i) Moderating effect of DIF

The regression models incorporate both DIF and the interaction term between NEU and DIF (cNEU×cDIF). Table 10 shows the results. E23 shows that the coefficient of cNEU×cDIF is 0.0025 ($p < 0.01$), which is significant, and E24 shows that the coefficient of cNEU×cDIF is 0.0026 ($p < 0.01$), which indicates that DIF has a positive moderating effect. The adj.$R^2$ in E23 and E24 are 0.903 and 0.891, respectively. The two models fit well. DIF has a complementary moderating effect, and the DIF moderating mechanism is strongly supported by the sample data. For every 1 unit increase in the moderating variable DIF, the marginal effect of NEU on GIN increases by 0.0025 units, and the marginal effect of NEU on GNS increases by 0.0026 units. The empirical results show that DIF has a significant positive moderating effect on NEU affecting urban GI, which not only amplifies the "quantity increase", but also enhances the "quality improvement" effect. The internal mechanism of this moderating effect is mainly as follows: DIF reduces the financing threshold, broadens the sources of funding, and injects more capital into the green technology R&D process, which meets the funding needs of GI development [67]. While alleviating financing constraints, DIF effectively reduces the matching friction of innovation elements between capital and technology, thereby enhancing the marginal contribution of NEU to urban GI. *Hypothesis 2* is verified.

#### ii) Moderating effect of DIF and GTS

Taking the median of GTS as the boundary, the samples are divided into two categories "strong government support cities" (GTS≥median) and "weak government support cities" (GTS<median). E25 shows that the coefficient of cNEU×cDIF is 0.0020 ($p < 0.01$), and E26 shows that the regression coefficient of cNEU×cDIF is 0.0015 ($p < 0.01$), which both meet the significance criterion. The adj.$R^2$ in E25 and E26 are 0.905 and 0.790, respectively. The two models fit well. DIF has a complementary moderating effect. In strong government support cities, for every unit increase in the moderating variable DIF, the marginal effect of NEU on GIN increases by 0.0020 units, while in weak government support cities, it increases

by 0.0015 units. The results show that the moderating effect of DIF has significant heterogeneity of policy environment. In cities with high level of government technological support, the promoting effect of DIF on NEU and urban GI "quantity increase" is more prominent. In cities with weak government technology support, the moderating role of DIF has not disappeared, but it is relatively weak. This shows that the improvement of government technology support can enhance the marginal strengthening effect of DIF in NEU promoting GIN. E27 shows the coefficient of cNEU × cDIF is 0.0021 ($p < 0.01$), and E28 shows the coefficient of cNEU × cDIF is 0.0012 ($p < 0.01$), which both meet the significance criterion. The results show that the moderating effect of DIF has significant policy environment heterogeneity, in cities with high level of government technology support, the promoting effect of DIF on NEU and urban GI "quality improvement" effect is more prominent, in cities with weak government technology support, the moderating role is relatively weak. This difference can be attributed to the role of government technology support in promoting the integration of industry and digital economy [68], specifically, government technology support not only improves the resource allocation efficiency of digital finance and reduces the marginal cost of financial services, but also accelerates the diffusion of green technology in the NEU process by alleviating the risk of enterprise R&D [69], thus significantly enhancing the marginal effect of DIF on the improvement of GI quality (Table 10).

### iii) Moderating effects of DIF and INT

Table 11 reports the results of the moderating effect analysis grouped by the median of informationization level (INT). According to the median of INT, the sample cities are divided into two groups: "high-informatization cities" (INT is higher than the median) and "low-informatization cities" (INT is lower than the median). E29 shows the coefficient of cNEU × cDIF is 0.0018 ($p < 0.01$), and E30 shows the coefficient of cNEU × cDIF is 0.0015 ($p < 0.01$), both of which are significant. The adj. $R^2$ in E29 and E30 are 0.921 and 0.882, respectively. The two models fit well. DIF has a complementary moderating effect. In high-information cities, for every unit increase in the moderating variable DIF, the marginal effect of NEU on GIN increases by 0.0018 units, while in low-information cities, it increases by 0.0015 units. The results show that the moderating effect of DIF is significantly stronger in high-informatization cities than in low-informatization cities. Although the

**Table 10. Moderating Effects Results I.**

| VAR | E23 | E24 | E25 | E26 | E27 | E28 |
|---|---|---|---|---|---|---|
| | full sample | | strong government support cities | weak government support cities | strong government support cities | weak government support cities |
| | GIN | GNS | GIN | GIN | GNS | GNS |
| NEU | 0.1524** (2.4531) | 0.1932*** (3.5573) | 0.2071** (2.3125) | 0.0123 (0.3269) | 0.2366*** (2.7229) | 0.0265 (0.9000) |
| DIF | 0.0020*** (3.2077) | 0.0019*** (3.3979) | 0.0037*** (5.5202) | −0.0002 (−0.6962) | 0.0031*** (4.5126) | 0.0001 (0.4965) |
| NEU × DIF | 0.0025*** (14.5822) | 0.0026*** (15.6078) | 0.0020*** (8.7446) | 0.0015*** (7.3472) | 0.0021*** (9.5349) | 0.0012*** (6.6281) |
| CV | Y | Y | Y | Y | Y | Y |
| cons | −0.2855* (−1.9410) | −0.2158 (−1.5652) | −0.0924 (−0.4414) | −0.3415*** (−4.2137) | −0.0551 (−0.2318) | −0.3742*** (−5.3851) |
| N | 3396 | 3396 | 1687 | 1683 | 1687 | 1683 |
| adj. $R^2$ | 0.903 | 0.891 | 0.905 | 0.790 | 0.886 | 0.744 |
| id FE | Y | Y | Y | Y | Y | Y |
| Year FE | Y | Y | Y | Y | Y | Y |

Note: See notes for Table 6

**Table 11. Moderating Effects Results II.**

| VAR | E29 | E30 | E31 | E32 |
|---|---|---|---|---|
| | high-informatization cities | low-informatization cities | high-informatization cities | low-informatization cities |
| | GIN | GIN | GNS | GNS |
| NEU | −0.0381 (−0.3596) | 0.1665*** (4.3957) | 0.0141 (0.1389) | 0.1567*** (3.9211) |
| DIF | 0.0017 (1.3353) | 0.0007*** (3.0680) | 0.0016 (1.2866) | 0.0009*** (3.3635) |
| NEU×DIF | 0.0018*** (5.9748) | 0.0015*** (6.6361) | 0.0019*** (6.6277) | 0.0016*** (7.0950) |
| CV | Y | Y | Y | Y |
| cons | 0.2247 (0.6233) | −0.4645*** (−7.8654) | 0.4011 (1.1564) | −0.4627*** (−6.7978) |
| N | 1698 | 1684 | 1698 | 1684 |
| adj. $R^2$ | 0.921 | 0.882 | 0.914 | 0.823 |
| id FE | Y | Y | Y | Y |
| Year FE | Y | Y | Y | Y |

Note: See notes for Table 6

moderating effect of DIF still exists in low-informatization cities, its intensity is relatively limited. This shows that in highly informatized cities, DIF has a stronger effect on NEU, thereby affecting GIN. E31 shows the coefficient of cNEU×cDIF is 0.0019 (p < 0.01), and E32 shows the coefficient of cNEU×cDIF is 0.0016 (p < 0.01), both of which passed the significance test. It indicates that the moderating effect of DIF is stronger in high-informatization cities and that DIF can better strengthen the effect of NEU on GNS. DIF relies on an informatization infrastructure, and informatization needs financial support [70]. In high-informatization cities, with the help of digital technology, it is possible to achieve more accurate and rapid matching between digital financial resources and green technology needs. Also, it can effectively reduce transaction costs, thereby promoting key production factors such as capital, technology, and talent to cluster towards the field of GI with higher efficiency, ultimately significantly strengthening the intrinsic influence between NEU and GI.

**iv) Moderating effects of DIF and AGDP**

Table 12 reports the regression results grouped by median per capita GDP (AGDP). According to the AGDP median, the sample cities were divided into two groups: developed cities (AGDP above the median) and underdeveloped cities (AGDP below the median). The moderating variables and their interactions with NEU are included in the regression model to test the heterogeneous moderating effect of NEU affecting GIN. The coefficient of cNEU×cDIF is 0.0018 (p < 0.01) in the "developed cities". The interaction term is 0.0011 (p < 0.01) in the "underdeveloped cities." The adj.$R^2$ in E33 and E34 are 0.898 and 0.750, respectively. The two models fit well. DIF has a complementary moderating effect. In developed cities, for every unit increase in the moderating variable DIF, the marginal effect of NEU on GIN increases by 0.0018 units, while in underdeveloped cities, it increases by 0.0011 units. It indicates that the moderating effect of DIF is more pronounced in "developed cities"; DIF better enhances the effect of NEU on GIN. DIF and cNEU×cDIF are incorporated into the E35 model. The coefficient of cNEU×cDIF is 0.0020 (p < 0.01) in the "developed cities" and 0.0010 (p < 0.01) in the "underdeveloped cities". The results show that the DIF's moderating effect is more pronounced in developed cities. This difference is due to the "digital-capital-market" synergy mechanism generated by the coupling of DIF and economic development level: on the one hand, DIF uses digital technology to achieve accurate matching of capital supply and demand, shortening the cycle of green technology from R&D to commercialization [71], on the other hand, the market scale advantage and

**Table 12. Moderating Effects Results III.**

| VAR | E33 | E34 | E35 | E36 |
|---|---|---|---|---|
| | developed cities | underdeveloped cities | developed cities | underdeveloped cities |
| | GIN | GIN | GNS | GNS |
| NEU | 0.2477*** (2.8601) | 0.0842** (2.0825) | 0.2677*** (3.2968) | 0.0843** (2.3639) |
| DIF | 0.0013 (1.6333) | 0.0011*** (4.8122) | 0.0014* (1.7867) | 0.0012*** (4.3486) |
| NEU × DIF | 0.0018*** (7.6070) | 0.0011*** (7.3311) | 0.0020*** (8.5370) | 0.0010*** (5.9910) |
| CV | Y | Y | Y | Y |
| cons | 0.5922** (2.5705) | −0.5913*** (−11.2765) | 0.5429** (2.1914) | −0.6113*** (−10.0551) |
| N | 1686 | 1682 | 1686 | 1682 |
| adj. $R^2$ | 0.898 | 0.750 | 0.882 | 0.640 |
| id FE | Y | Y | Y | Y |
| Year FE | Y | Y | Y | Y |

Note: See notes for Table 6

technology absorption capacity of developed cities make the innovation achievements spread rapidly and produce economies of scale. The superposition of the two significantly magnifies the positive impact of NEU on urban GI.

### v) Moderating effects of DIF and ENC

Different cities exhibit significant differences in industrial structure, resource endowment, technical level and energy policy, leading to differences in energy consumption and energy efficiency. Based on the median split of ENC, cities are categorized into two groups: "high-energy-consuming cities" (ENC≥median) and "low-energy-consuming cities" (ENC<median). The coefficient of cNEU × cDIF is 0.0022 (p < 0.01) in "high-energy-consuming cities" and 0.0018 (p < 0.01) in the "low-energy-consuming cities" (Table 13). DIF has a complementary moderating effect. In high-energy-consuming cities, for every unit increase in the moderating variable DIF, the marginal effect of NEU on GIN increases by 0.0022 units, while in low-energy-consuming cities, it increases by 0.0018 units. This supports that the moderating effect of DIF is stronger in "high-energy-consuming cities". The coefficient of cNEU × cDIF is 0.0024 (p < 0.01) in "high-energy-consuming cities" and 0.0018 (p < 0.01) in "low-energy-consuming cities". The results showed that the moderating effect of DIF is more prominent in the "high-energy-consuming cities". This difference is due to the interaction between DIF and ENC: DIF provides capital guarantee for green technology research and development and achievement transformation in high energy-consuming fields by optimizing the allocation of funds; on the other hand, environmental pressures and policy constraints caused by high energy consumption have significantly increased the intensity of urban demand for GI [72]. The superposition of the two forms a positive feedback loop of "demand pull-fund supply", which makes the driving effect of NEU on GNS significantly amplified in high-energy-consuming cities.

In summary, Hypothesis 3 is verified.

## 5. Further analysis

To test for non-linearity in the effect of NEU on GI, the following panel threshold model is established:

**Table 13. Moderating Effects Results IV.**

| VAR | E37 | E38 | E39 | E40 |
|---|---|---|---|---|
| | high-energy-consuming cities | low-energy-consuming cities | high-energy-consuming cities | low-energy-consuming cities |
| | GIN | GIN | GNS | GNS |
| NEU | 0.1852** (2.4201) | 0.0829 (1.0366) | 0.2294*** (3.1698) | 0.0580 (1.0440) |
| DIF | 0.0046*** (6.4742) | 0.0002 (0.4283) | 0.0038*** (5.5517) | 0.0008 (1.6416) |
| NEU×DIF | 0.0022*** (10.4745) | 0.0018*** (8.8371) | 0.0024*** (11.4880) | 0.0018*** (8.4434) |
| CV | Y | Y | Y | Y |
| cons | −0.3958** (−2.1063) | −0.4096*** (−3.2735) | −0.2392 (−1.1678) | −0.4892*** (−3.9769) |
| N | 1676 | 1681 | 1676 | 1681 |
| adj. $R^2$ | 0.908 | 0.848 | 0.893 | 0.803 |
| id FE | Y | Y | Y | Y |
| Year FE | Y | Y | Y | Y |

Note: See notes for Table 6

$$GIN_{it} = \alpha_{20} + \rho_0 NEU_{it} \cdot I\left(GES_{it} \leq \omega_1\right) + \rho_1 NEU_{it} \cdot I\left(\omega_1 < GES_{it} \leq \omega_2\right) + \cdots + \rho_m NEU_{it} \cdot I\left(\omega_{m-1} < GES_{it} \leq \omega_m\right)$$
$$+ \rho_{m+1} NEU_{it} \cdot I\left(GES_{it} > \omega_m\right) + \gamma_{20} CV_{it} + \mu_i + \delta_t + \varepsilon_{it} \tag{18}$$

$$GNS_{it} = \alpha_{21} + \tau_0 NEU_{it} \cdot I\left(GES_{it} \leq \epsilon_1\right) + \tau_1 NEU_{it} \cdot I\left(\epsilon_1 < GES_{it} \leq \epsilon_2\right) + \cdots + \tau_m NEU_{it} \cdot I\left(\epsilon_{m-1} < GES_{it} \leq \epsilon_m\right)$$
$$+ \tau_{m+1} NEU_{it} \cdot I\left(GES_{it} > \epsilon_m\right) + \gamma_{21} CV_{it} + \mu_i + \delta_t + \varepsilon_{it} \tag{19}$$

$\rho$ is the coefficient of NEU, $I(\cdot)$ is an indicator function, $\omega$ and $\epsilon$ are threshold values.

The threshold variable is Government's green support (GES), measured as the ratio of fiscal environmental protection expenditure to fiscal general budget expenditure. Bootstrap testing indicates that, regardless of whether the dependent variable is GIN or GNS, only the single threshold effect is significant, while the double and triple threshold effects are not. This indicates that the impact of GES on both GIN and GNS exhibits a single threshold effect, with a threshold value of 0.0104.

This paper further examines the non-linear impact of NEU on GIN and GNS under the GES framework. Table 14 shows that when GES ≤ 0.0104, the coefficient of NEU on GIN is 0.6770, which is significant. When GES > 0.0104, the coefficient of NEU on GNS is 0.7866, which is significant. It means that greater government spending on environmental protection amplifies the impact of NEU on GIN, driving the "quantity increase" effect toward a trajectory of rising marginal effect. From the perspective of factor supply, when the government's support for green industries is low, green R&D projects will be constrained by funding and policy support [73]. As the government's support for green industries gradually increases, GI's financial constraints are alleviated. When NEU needs a lot of green infrastructure and technology, financial support helps address funding shortfalls, making it easier for urbanization to drive the "quantity increase" of GI. Moreover, the government's support behavior will also play a guiding role in green development and help inhibit pollution, thereby limiting high-emission, high-energy-consuming industries and promoting new formats such as pollution-monitoring equipment and solid waste resource utilization [74]. Therefore, as the government's green support increases, it helps facilitate NEU's "quantity increase" effect on GI.

**Table 14. Threshold Effect Estimation Results.**

| VAR | E41 | VAR | E42 |
|---|---|---|---|
| | GIN | | GNS |
| CV | Y | CV | Y |
| NEU(GES ≤ 0.0104) | 0.6770*** (5.8241) | NEU(GES ≤ 0.0104) | 0.7191*** (5.5931) |
| NEU(GES > 0.0104) | 0.7866*** (6.0522) | NEU(GES > 0.0104) | 0.8317*** (5.9532) |
| cons | 0.2598** (2.3312) | cons | 0.3454*** (2.7242) |
| N | 3396 | N | 3396 |
| adj. $R^2$ | 0.4952 | adj. $R^2$ | 0.4643 |
| id FE | Y | id FE | Y |
| Year FE | Y | Year FE | Y |

Note: See notes for Table 6

Similarly, when GES ≤ 0.0104, the coefficient of NEU on GNS is 0.7191, which is significant. When GES > 0.0104, the coefficient of NEU on the GNS is 0.8317, which is significant. The "quality improvement" effect of GI often requires policy support and continuous investment. When the GES is low, the enterprise's investment in green R&D depends on the enterprise itself, which affects continuous R&D investment, and the GI's "quality improvement" is limited. When the GES reaches a high level, government funds can support enterprises to carry out long-term technology iteration and result verification, so that enterprises are willing to invest more time and resources in "quality improvement" to optimize the stock technology and transform the process flow, and lead it to high efficiency and green transformation [75], so as to improve the technical level and practical value, and promote more high-quality GI results. Moreover, from the perspective of technology diffusion, when GES is high, enterprises can participate in cross-regional and cross-industry exchanges and cooperation, which is more conducive to promoting technological exchanges to achieve deep knowledge integration and re-innovation [76], promote green technology from follow-up innovation to leading innovation, and the quality of innovation results has been significantly improved. Therefore, with the increase of government green support, it is beneficial to expand the "quality improvement" effect of NEU on urban GI.

## 6. Conclusion and Policy Implications

### 6.1. Summary

During the 14th Five-Year Plan period, China's urbanization is still in a stage of rapid development, with both opportunities and challenges. In this context, the NEU strategy promotes the transformation of urban development mode, constructs a new pattern, enhances urban innovation ability and performance, and promotes sustainable economic development by coordinating resources and factors. This paper empirically analyzes the "quantity increase" and "quality improvement" effects of NEU on urban GI by adopting urban panel data from 2011 to 2022. The analysis suggests that NEU acts as a catalyst for GI, positively fostering both quantity increases and quality improvements. Robustness and endogeneity checks confirm the reliability of the findings. DIF reinforces the positive influence of NEU on urban GI. The positive moderating role of DIF on the impact of NEU on GIN and GNS is significantly strengthened by higher levels of government technology support, informatization, regional economic development, and energy consumption. The impact of NEU on urban GI exhibits a nonlinear relationship, and the stronger the government's green support, the more conducive it is for NEU to exert "quantity increase" and "quality improvement" effects on urban GI. Our research not only confirms previous studies' findings but also extends them with concrete evidence. DIF significantly amplifies NEU's positive impact on GI, offering a

new perspective and expanding DIF's application in GI. This multi-factor interaction analysis provides a more comprehensive understanding of NEU and GI and extends on previous single-factor or limited-factor studies. GES plays a crucial role in enhancing the impact of NEU on GIN and GNS, which confirms the overall positive impact of government support on innovation by exploring the specific role of GES in the relationship between NEU and GI.

## 6.2. Theoretical and policy implications

Our research focuses on the impact of NEU on urban GI, challenging the traditional view that urbanization prioritizes economic growth and resource use. NEU emphasizes green development and ecological benefits, and its role in promoting GI is more evident, which aligns with the new urban development theory. It provides a new theoretical perspective for how urbanization promotes GI and enriches the existing theory of the relationship between urbanization and innovation. This paper discusses the "quantity increase" and "quality improvement" effects of NEU on GI, expands the theoretical framework of GI research from one-dimensional to two-dimensional, and enriches the research results of NEU. DIF amplifies the effect of NEU on urban GI, revealing its nonlinear characteristics and multivariate synergistic regulation mechanism (digital inclusive finance, government technology support, informatization, regional economy, energy consumption). This not only deepens the Porter hypothesis and endogenous growth theory in the context of green transformation but also helps reveal the internal logic and complex mechanisms by which NEU affects urban GI more comprehensively and deeply. It also provides a new theoretical explanation dimension for understanding how the institutional, financial and technological environment jointly shape the impact path of urbanization on innovation. This study not only provides policy reference for the construction of NEU, but also provides important empirical evidence and a new analytical perspective for the development of urbanization-GI theory with the characteristics of "transitional economy" and provides new evidence for verifying and expanding international research on the interaction mechanism of urbanization, digital finance and GI. Compared with relevant international research, existing research mainly analyzes how NEU can promote urban GI from the perspectives of enterprises [77], policy [78,79], and green development [80]. However, urbanization models differ across regions, and their impacts on regional green technology R&D also vary [8]. Moreover, the impact of NEU on GI also needs to be mediated by variables. For example, NEU promotes GI by alleviating corporate financial constraints [9]. In addition, existing international research often treats NEU as an independent driver in the research innovation system. This study not only analyzes the "quantity increase" and "quality improvement" effects of NEU and urban GI, but also incorporates NEU with DIF, GTS, INT, AGDP, and ENC into the same analytical framework, revealing the complex mechanism, which provides a new perspective for understanding the relationship between NEU and GI in different contexts.

Therefore, we need to leverage NEU's dual role in promoting GI, scientifically plan urban functional areas, rationally arrange green industries and high-energy-consuming industries, and build specialized green industrial agglomeration areas to provide space for GI. Combined with NEU's quality-improvement effect on GI, it guides the population to gather in green industries, encourages enterprises to pursue green product innovation, and transforms the population's urbanization advantage into a talent-and-technology advantage in urban GI. Focusing on the synergistic effects of DIF, government science and technology support, informatization and other factors, this paper constructs a digital service platform for green finance, integrates green technology demand and financing information, develops appropriate green credit products and provides accurate financial support for enterprises and scientific research institutions. Explore the establishment of a cross-sectoral information sharing mechanism, promote the cross-regional optimal allocation of financial resources, and amplify the linkage dividend between NEU and green finance. In view of the strengthening effect of government green support on innovation effect, it is suggested to maintain a reasonable proportion of fiscal expenditure on environmental protection, reduce the financing cost of green projects through loan subsidies and guiding funds, attract social capital to participate in GI, and provide policy guarantee for NEU to release greater innovation efficiency.

 

### 6.3. Limitations and future research

It should be noted that while we use green patent applications and green invention patent applications as measures of the "quantity increase" and "quality improvement" of urban GI, respectively, relying solely on patent data as a proxy for all GI has several potential limitations. Because not all innovations are patented, it is challenging to capture the full scope of GI using patent data. In addition, although we have employed various methods in the paper to test the robustness and endogeneity of the analysis results, to alleviate concerns about the reliability of specific data proxies, data and method limitations still exist. China has a distinctive social system and urbanization process, and this study is mainly based on the analysis of China's urban panel data, which will inevitably affect the applicability of the research conclusions to other countries.

To further overcome these limitations, patent data can be combined with other data in future research on GI, and environmental performance indicators, energy efficiency, etc., can be included in GI analyses to more comprehensively cover its scope. In terms of research methods, future research should use spatial econometric methods or more rigorous causal inference methods to further verify the causal relationship between NEU and urban GI, incorporating qualitative research methods or quasi-experimental designs to provide stronger causal evidence. Future research can also further expand the scope of the study, analyze the relationship between urbanization and urban GI in other countries, and compare it with China to test the boundary conditions of our research results.

## Author contributions

**Conceptualization:** Liang Fang, Chengxiang Li.

**Formal analysis:** Liang Fang, Chen Jin.

**Funding acquisition:** Liang Fang.

**Investigation:** Liang Fang, Chengxiang Li.

**Methodology:** Liang Fang, Chengxiang Li.

**Project administration:** Liang Fang, Chen Jin.

**Resources:** Liang Fang.

**Supervision:** Liang Fang, Chengxiang Li.

**Validation:** Liang Fang, Chen Jin.

**Visualization:** Liang Fang, Chen Jin.

**Writing – original draft:** Liang Fang, Chengxiang Li.

**Writing – review & editing:** Liang Fang.

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
