## [Decision Letter · Decision Letter 0]

10 Sep 2025

PONE-D-25-37471The Impact of New Urbanization on Quantity Increase and Quality Improvement of Green Innovation: Based on Multivariate Moderating Effect ModelsPLOS ONE

Dear Dr. Fang,

Thank you for submitting your manuscript to PLOS ONE. After careful consideration, we feel that it has merit but does not fully meet PLOS ONE’s publication criteria as it currently stands. Therefore, we invite you to submit a revised version of the manuscript that addresses the points raised during the review process.

We look forward to receiving your revised manuscript.

Kind regards,

Alejandro Botero Carvajal, Ph.D

Academic Editor

PLOS ONE

Journal Requirements:

2, Thank you for stating the following financial disclosure: [This study is supported by Humanities and Social Sciences Research Project of the Ministry of Education (20YJCZH028), Anhui Province Social Sciences In-novation and Development Research Project (2021CX047), Anhui Province Outstanding Talents Cultivation Project of Universities (gxbjZD2022062), Huangshan University Foundation Cultivation Project (2021GJYY006), and Huangshan University Research and Innovation Team (2021XCXTDPY04).].

3. Thank you for uploading your study's underlying data set. Unfortunately, the repository you have noted in your Data Availability statement does not qualify as an acceptable data repository according to PLOS's standards.

Reviewers' comments:

Reviewer's Responses to Questions

**Comments to the Author**

1. Is the manuscript technically sound, and do the data support the conclusions?

Reviewer #1: Partly

Reviewer #2: Yes

Reviewer #3: Yes

2. Has the statistical analysis been performed appropriately and rigorously? 

Reviewer #1: Yes

Reviewer #2: Yes

Reviewer #3: Yes

3. Have the authors made all data underlying the findings in their manuscript fully available?

Reviewer #1: No

Reviewer #2: Yes

Reviewer #3: Yes

4. Is the manuscript presented in an intelligible fashion and written in standard English?

Reviewer #1: Yes

Reviewer #2: Yes

Reviewer #3: Yes

5. Review Comments to the Author

Reviewer #1: Dear authors;

This article explores the urban development program focusing on green innovations, which is noteworthy in itself.

This article investigates the urban development program concerning green innovations, which is intriguing in its own way.

This article analyzes the urban development initiative aimed at green innovations, which is captivating on its own.

The analytical approach is also grounded in statistical models;

The methodology of analysis is likewise rooted in statistical models;

which can be accomplished using historical data.

which can be executed based on historical data.

which can be achieved on the foundations of historical data.

Considering the extensive amount of calculations, several points need clarification:

Regarding the significant volume of calculations, some aspects must be elucidated:

Where are the analyzed data sourced from?

Reviewer #2: The paper addresses a highly relevant and timely topic, especially in the context of China's national strategy for sustainable development. The methodology employed is robust, including various econometric models and comprehensive checks for robustness and endogeneity. The findings offer significant policy implications, making a valuable contribution to the literature. The minors weaknesses in need of strengthening are indicated below:

1. Even though the paper uses green patent applications (GIN) and green invention patent applications (GNS) as measures for the "quantity increase" and "quality improvement" of GI respectively , it would be beneficial to briefly acknowledge the potential limitations of relying solely on patent data as a proxy for all green innovation (e.g., not all innovations are patented, varying implementation rates, or challenges in capturing the full scope of GI).

2. The current "Discussion" section (Section 6) functions primarily as a "Summary" and "Recommendations" . To maximize the paper's impact, a dedicated "Discussion" subsection could be expanded to:

o More explicitly contextualize the findings within the broader literature, highlighting how they extend, confirm, or diverge from previous research.

o Elaborate on the theoretical implications of the findings.

o Discuss potential limitations of the study (e.g., generalizability beyond China, reliance on specific data proxies) that are not already covered by the robustness checks.

3. While the English is generally clear, there are a few instances of phrasing that could be slightly refined for improved conciseness and flow (e.g., "amplify the dividends of new urbanization and exert the GI effect"; "the regression results are resilient" ; "transformation from quantity to quality" ). A final proofread by a native English speaker could further polish the manuscript.

Reviewer #3: This paper analyzes whether China’s new urbanization (NEU) drives both the quantity and the quality of green innovation. Manuscript satisfactory. However, standardize acronym use, by defining NEU, GI, DIF, GTS, INT, AGDP, ENC, GES once, then use consistently

6. PLOS authors have the option to publish the peer review history of their article (what does this mean?). If published, this will include your full peer review and any attached files.

Reviewer #1: No

Reviewer #2: **Yes:**Michael Addaney

Reviewer #3: No

---

## [Author Response · Author response to Decision Letter 1]

24 Sep 2025

(Manuscript Number: PONE-D-25-37471)

Dear editor and reviewers,

Thank you for your letter and for the reviewers’ valuable and helpful comments on our manuscript entitled “The impact of new urbanization on quantity increase and quality improvement of green innovation: based on multivariate moderating effect models” (ID: PONE-D-25-37471). We have carefully read all comments and have made corresponding corrections. We hope that our revisions have significantly improved the quality of the paper. The modified content has been marked to highlight the changes. The responses to the reviewers’ comments are as follows.

Response to Reviewer #1 Comments

Reviewer #1:

This article explores the urban development program focusing on green innovations, which is noteworthy in itself. This article investigates the urban development program concerning green innovations, which is intriguing in its own way. This article analyzes the urban development initiative aimed at green innovations, which is captivating on its own. The analytical approach is also grounded in statistical models;

The methodology of analysis is likewise rooted in statistical models;

which can be accomplished using historical data.

which can be executed based on historical data.

which can be achieved on the foundations of historical data.

Response:

We sincerely appreciate the reviewer's thoughtful evaluation and insightful comments on our manuscript, particularly the recognition of the value in our analysis of the urban development program concerning green innovations. This validates the significance of our research focus, strengthens our confidence in exploring the interdisciplinary nexus between sustainability and urban planning, and motivates us to refine our methodologies to enhance both academic rigor and practical applicability.

Comment 1:

Regarding the significant volume of calculations, some aspects must be elucidated:

Where are the analyzed data sourced from?

Response:

We sincerely appreciate the reviewer for emphasizing the critical need to clarify the data sources on which our analysis is based. We fully acknowledge that declaring the source of data is essential for analyzing data and validating results. However, we regret that this aspect was not adequately addressed in the original submission. To clarify the specific sources of the data more explicitly, we have added a new section titled "3.3 Data sources" in Part 3 (3. Research design) of the paper. In this section, we have provided detailed information on the data sources for each variable, including specific databases, government reports, and public datasets. The specific details are outlined below:

This paper utilizes panel data from 284 cities spanning the period from 2011 to 2022. The data for GIN and GNS are from the Chinese Research Data Services Platform (CNRDS), NEU encompasses multiple second-level indicators, and the data for these indicators mainly originate from the China Urban Statistical Yearbook, statistical yearbooks of various regions, the EPS database and the China Energy Statistical Yearbook. The data for control variables FDI, TEL, OOW, HCL, RFV and SCL are from the China Urban Statistical Yearbook, and the data for INT is from the EPS database. The data for DIF is sourced from the Digital Financial Inclusion Index published by Peking University. The data for moderating variables GTS, INT, AGDP are from the China Urban Statistical Yearbook, and the data for ENC is from the China Energy Statistical Yearbook. The data for GES is from the zldatas database.

Response to Reviewer #2 Comments

Reviewer #2:

The paper addresses a highly relevant and timely topic, especially in the context of China's national strategy for sustainable development. The methodology employed is robust, including various econometric models and comprehensive checks for robustness and endogeneity. The findings offer significant policy implications, making a valuable contribution to the literature.

Response:

We sincerely appreciate the reviewer's thorough review and insightful comments provided on our manuscript. The comments reassure us that our analytical framework resonates with scholarly standards in applied economics. The comments also motivate us to refine these models further in future research.

Comment 1:

1. Even though the paper uses green patent applications (GIN) and green invention patent applications (GNS) as measures for the "quantity increase" and "quality improvement" of GI respectively , it would be beneficial to briefly acknowledge the potential limitations of relying solely on patent data as a proxy for all green innovation (e.g., not all innovations are patented, varying implementation rates, or challenges in capturing the full scope of GI).

Response:

We thank the reviewer for the valuable comments and we completely agree that relying solely on patent data as a proxy for all green innovation has potential limitations. As the reviewer mentioned, not all innovations are patented, and there are varying implementation rates among different innovations. Moreover, it is indeed challenging to capture the full scope of green innovation using only patent data. We have made modifications to the paper according to the reviewer’s suggestion. In the relevant section where we have introduced the use of green patent applications and green invention patent applications as measures, we have added a discussion on these limitations. We have elaborated on how these limitations may affect the accuracy and comprehensiveness of our research results, and also discussed possible ways to mitigate these limitations in future research. We have added the content of “6.3. Limitations and future research” in the last section “6. Discussion” of the paper. We believe that adding this discussion will make our research more rigorous and comprehensive and enhance the credibility of our findings. Specifically, the following content has been added:

It should be pointed out that while we use green patent applications and green invention patent applications as measures for the "quantity increase" of GI and "quality improvement" of GI, respectively, relying solely on patent data as a proxy for all GI has several potential limitations. For example, the level of GI may have been underestimated; not all innovations are patented, the implementation rates of patented GI can vary significantly, and not all green patents can be applied in practice. It is challenging to capture the full scope of GI using patent data. In order to further overcome these limitations, patent data can be combined with other data in future research on GI, and environmental performance indicators, energy efficiency, etc., can be included in the analysis of GI to more comprehensively cover the scope of GI.

Comment 2:

2. The current "Discussion" section (Section 6) functions primarily as a "Summary" and "Recommendations" . To maximize the paper's impact, a dedicated "Discussion" subsection could be expanded to:

o More explicitly contextualize the findings within the broader literature, highlighting how they extend, confirm, or diverge from previous research.

o Elaborate on the theoretical implications of the findings.

o Discuss potential limitations of the study (e.g., generalizability beyond China, reliance on specific data proxies) that are not already covered by the robustness checks.

Response:

We would like to thank the reviewer for the thoughtful and constructive suggestions. We have carefully considered each of the comments and made significant revisions. We indeed should further expand the content of the "Discussion" section (Section 6). We have revised the 'Discussion' section of the paper to better align our findings with existing literature. The following is a detailed description of our modifications:

(1) We have integrated the summarized content into the discussion section (Section 6) by comparing and discussing existing literature to clearly demonstrate the relationship between our research and previous studies. The following specific content has been added to the "6.1. Summary" section:

Our research provides detailed empirical evidence through empirical analysis using urban panel data. It not only confirms the viewpoints of previous studies but also expands upon them by providing concrete evidence. DIF significantly amplifies the positive impact of NEU on GI, which is a new perspective and expands the application of DIF in the field of GI. This multi-factor interaction analysis provides a more comprehensive understanding of NEU and GI and extends previous single-factor or limited-factor studies. GES plays a crucial role in enhancing the impact of NEU on GIN and GNS, which confirms the overall positive impact of government support on innovation and expands research by exploring the specific role of GES in the relationship between NEU and GI.

(2) We have expanded the content of theoretical implications in the Discussion section (6. Discussion) by adding a new subsection "6.2. Theoretical and policy implications" to provide a more in-depth analysis and elaborate on the theoretical contribution of this study. The following specific content has been added:

Our research focuses on the impact of NEU on GI, challenging the traditional view that urbanization mainly focuses on economic growth and resource utilization. NEU emphasizes green development and ecological benefits, and has a more obvious promoting effect on GI. It is consistent with the viewpoint of the new urban development theory, providing a new theoretical perspective on how urbanization can promote GI and enriching existing theories on the relationship between urbanization and innovation. DIF amplifying the impact of NEU on GI indicates the need to further strengthen attention to the theory of enhancing financial inclusiveness and industrial integration, which opens up a new path for the integration of digital technology innovation and green development concepts into urban management and planning. The study of the interaction of multiple factors emphasizes the importance of multi-factor interaction in innovation theory, which provides a more comprehensive understanding of innovation theory and helps to formulate more effective policies to promote GI.

(3) We have also added new content in section “6.3. Limitations and future research", the potential limitations are discussed in detail, and directions for future research to address these issues are proposed. The following specific content has been added:

China has a distinctive social system and urbanization process, and this study is mainly based on the analysis of China's urban panel data, which will inevitably affect the applicability of the research conclusions to other countries. Therefore, future research can further expand the scope of the study, analyze the relationship between urbanization and GI in other countries, and compare it with China to test the boundary conditions of our research results. In addition, although we have employed various methods in the paper to test the robustness and endogeneity of the analysis results, in order to alleviate concerns about reliability on specific data proxies, data limitations still exist. Future research can adopt various research methods, such as incorporating qualitative research methods or adopting quasi-experimental research methods, to provide more causal evidence for the relationship between NEU and GI.

Comment 3:

3. While the English is generally clear, there are a few instances of phrasing that could be slightly refined for improved conciseness and flow (e.g., "amplify the dividends of new urbanization and exert the GI effect"; "the regression results are resilient" ; "transformation from quantity to quality" ). A final proofread by a native English speaker could further polish the manuscript.

Response:

We would like to apologize to the reviewer for the language used in our manuscript. Perhaps due to repetitive content modifications, there are some language issues. We have invited professionals in the field of English language to examine the language used in our paper, discussed each specific example mentioned, and provided a revised version. We really hope to have a substantial improvement in language proficiency. The modifications made to some phrases are presented as follows, some modifications to other statements and phrases have been marked in the paper.

Original phrase: "amplify the dividends of new urbanization and exert the GI effect"

Revised phrase: “unleash the development dividends brought by NEU and fully leverage the positive effects of GI.”

Original phrase: "the regression results are resilient"

Revised phrase: "the regression results exhibit robustness"

Original phrase: NEU in China is in the development stage of "quantity is stronger than quality" and "transformation from quantity to quality".

Revised phrase: NEU in China is in a transitional phase from "quantity-based growth" to "quality-oriented enhancement".

Original phrase: “NEU is playing an increasingly significant role in GI through different ways, such as talent pooling, consumption upgrading, and development mode.”

Revised phrase: “At the critical stage of current social development, NEU, as a systematic and comprehensive strategy, is exerting increasingly significant and far-reaching effects on GI from different dimensions, effectively promoting the rapid development and efficient application of GI in various fields.”

Original phrase: “the regression coefficients are unchanged, which confirms the reliability of the benchmark regression”

Revised phrase: “there is no obvious alterations in the positive/negative orientations or quantitative value of the coefficient, which verifies the validation for the results”

Original phrase: “high informatization cities can more efficiently match digital financial resources with green technology needs, reduce transaction costs, capital, technology, and talent can flow more efficiently to the field of GI, which strengthening the connection between NEU and GI.”

Revised phrase: “in high informatization cities, with their advanced digital infrastructure and efficient information dissemination mechanisms, it is possible to achieve more accurate and rapid matching between digital financial resources and green technology needs. This precise matching effectively reduces transaction costs, thereby promoting key production factors such as capital, technology, and talent to cluster towards the field of GI with higher efficiency, ultimately significantly strengthening the intrinsic influence between NEU and GI.”

Original phrase: “amplifies the dividends of NEU and exert strong "quantity increase" and "quality improvement" effects on GI.”

Revised phrase: “fully unleash the development dividends brought by NEU and exert strong "quantity increase" and "quality improvement" effects of GI”

Response to Reviewer #3 Comments

Reviewer #3:

This paper analyzes whether China’s new urbanization (NEU) drives both the quantity and the quality of green innovation. Manuscript satisfactory.

Response:

We sincerely thank the reviewer's thoughtful evaluation and insightful comments on our manuscript. The reviewer's comments boost our confidence and enthusiasm for this research. We'll use this motivation to revise the paper further and try our best to improve its quality.

Comment 1:

However, standardize acronym use, by defining NEU, GI, DIF, GTS, INT, AGDP, ENC, GES once, then use consistently.

Response:

We sincerely appreciate the reviewer’s attention to this critical detail of consistent use of acronyms in our manuscript. We fully agree that standardized terminology can improve clarity and readability. As suggested, we have now defined all acronyms when they first appear in the text, marked their abbreviations in parentheses, and ensured their consistent use throughout the manuscript. We have also checked the entire document to eliminate any inconsistent or inexplicable abbreviations. The definition of each acronym is as follows:

GI: green innovation

NEU: new urbanization

GIN: “quantity increase” of GI

GNS: "quality improvement" of GI

CV: control variables

FDI: foreign direct investment

TEL: technology investment level

OOW: opening to the outside

HCL: human capital level

INT: the Internet pene

---

## [Decision Letter · Decision Letter 1]

21 Oct 2025

PONE-D-25-37471R1The Impact of new urbanization on quantity increase and quality improvement of green innovation: Based on multivariate moderating effect modelsPLOS ONE

Dear Dr. Fang,

Thank you for submitting your manuscript to PLOS ONE. After careful consideration, we feel that it has merit but does not fully meet PLOS ONE’s publication criteria as it currently stands. Therefore, we invite you to submit a revised version of the manuscript that addresses the points raised during the review process.

We look forward to receiving your revised manuscript.

Kind regards,

Alejandro Botero Carvajal, Ph.D

Academic Editor

PLOS ONE

Journal Requirements:

Reviewers' comments:

Reviewer's Responses to Questions

**Comments to the Author**

1. If the authors have adequately addressed your comments raised in a previous round of review and you feel that this manuscript is now acceptable for publication, you may indicate that here to bypass the “Comments to the Author” section, enter your conflict of interest statement in the “Confidential to Editor” section, and submit your "Accept" recommendation.

Reviewer #1: All comments have been addressed

Reviewer #2: All comments have been addressed

2. Is the manuscript technically sound, and do the data support the conclusions?

Reviewer #1: Yes

Reviewer #2: Yes

3. Has the statistical analysis been performed appropriately and rigorously? 

Reviewer #1: Yes

Reviewer #2: Yes

4. Have the authors made all data underlying the findings in their manuscript fully available?

Reviewer #1: No

Reviewer #2: Yes

5. Is the manuscript presented in an intelligible fashion and written in standard English?

Reviewer #1: Yes

Reviewer #2: Yes

6. Review Comments to the Author

Reviewer #1: Dear Authors,

After reviewing your manuscript, I have some technical suggestions to improve its quality, ensuring the availability of the necessary data:

It is recommended to use exploratory or confirmatory factor analysis, and possibly both, on the data. Additionally, principal component analysis can be beneficial for understanding the relationships among independent variables and the degree of their interconnections, leading to deeper and more advanced conclusions. Furthermore, utilizing residual plots against fitted values or error-related indices for evaluating regression results, especially in equations with high R² values, can enhance the quality of the manuscript's conclusions.

I am concerned about the lack of understanding of the relationships among the independent variables in this manuscript. Using Cronbach's alpha to demonstrate these relationships, along with presenting a fitted distribution curve for the data, seems essential for better understanding the nature of the data.

Reviewer #2: This revised manuscript addresses a highly relevant and timely topic concerning the relationship between New Urbanization (NEU) and Green Innovation (GI) in the context of China's national development strategy. The study uses empirical analysis based on panel data from 284 cities spanning 2011 to 2022. The methodology appears strong, incorporating various econometric models, including fixed effects, robustness checks (removing special city/year samples, replacing variables), and instrumental variable methods (LAG_NEU and NLV) to address endogeneity concerns. The primary findings that NEU positively contributes to both the quantity increase (GIN) and quality improvement (GNS) of GI, and that Digital Inclusive Finance (DIF) significantly amplifies this impact offer significant policy implications and make a valuable contribution to the literature. Importantly, the authors are commended for making substantial revisions in response to the previous reviewers' comments, particularly regarding data transparency, discussion depth, and linguistic clarity. These revisions have significantly improved the quality and rigor of the manuscript.

7. PLOS authors have the option to publish the peer review history of their article (what does this mean?). If published, this will include your full peer review and any attached files.

Reviewer #1: No

Reviewer #2: **Yes:**Michael Addaney

---

## [Author Response · Author response to Decision Letter 2]

7 Nov 2025

Response to Reviewers’ Comments

(Manuscript Number: PONE-D-25-37471R1)

Dear Reviewers,

We would like to express our heartfelt appreciation to the reviewers for your meticulous evaluation and insightful technical suggestions which have significantly strengthened our manuscript entitled “The impact of new urbanization on quantity increase and quality improvement of green innovation: based on multivariate moderating effect models” (ID: PONE-D-25-37471R1). Below is our point-by-point response to the comments:

Response to Reviewer #1 Comments

Reviewer #1:

After reviewing your manuscript, I have some technical suggestions to improve its quality, ensuring the availability of the necessary data:

It is recommended to use exploratory or confirmatory factor analysis, and possibly both, on the data. Additionally, principal component analysis can be beneficial for understanding the relationships among independent variables and the degree of their interconnections, leading to deeper and more advanced conclusions. Furthermore, utilizing residual plots against fitted values or error-related indices for evaluating regression results, especially in equations with high R² values, can enhance the quality of the manuscript's conclusions.

I am concerned about the lack of understanding of the relationships among the independent variables in this manuscript. Using Cronbach's alpha to demonstrate these relationships, along with presenting a fitted distribution curve for the data, seems essential for better understanding the nature of the data.

Response:

We sincerely thank the reviewer for the constructive feedback and detailed technical suggestions. Your technical suggestions are excellent and will undoubtedly significantly improve the quality of our work. We have carefully considered all your points and have developed a detailed revision plan to address them comprehensively. The key revisions we undertake are as follows:

1. This paper employs Exploratory Factor Analysis (EFA) and principal component analysis (PCA) to further examine the relationships and intercorrelations among the independent variable data. First, Cronbach’s alpha values are calculated for each indicator to assess data reliability. Then, the Kaiser-Meyer-Olkin (KMO) measure and Bartlett’s test of sphericity are conducted to evaluate the partial correlations among variables and the correlation matrix. All statistical measures meet the required criteria for the tests. Furthermore, as a part of the robustness checks, this study recalculated the explanatory variable NEU by replacing the composite score derived from the entropy weight method with a factor composite score obtained through Exploratory Factor Analysis (EFA). This approach is implemented in section "iv) Replacing independent variable" under "4.2 Robustness test" to verify the stability of the regression results.

Specific tests and indicators have been added to the section "ii) Independent variable" under "3.1. Indicator design" of the paper. The details are as follows:

In order to verify that there is a good reliability and correlation between the third-level indicators of NEU. Firstly, the reliability of the data is checked. The reliability of the data is measured by the Cronbach's alpha coefficient, and the Cronbach's alpha value is calculated for each of the three-level metrics as well as for all the metrics, and the results are obtained as shown in Table 2. The Cronbach's alpha value of each metric is greater than 0.7, and the Cronbach's alpha value of the overall reliability of the data is also greater than 0.7, which shows that the data satisfy the reliability requirements. Secondly, KMO and Bartlett's spherical test are utilized to assess the partial correlation and correlation matrix between the variables, as shown in Table 3. The result of the test is overall value of KMO is 0.7792>0.6, Bartlett of sphericity passes the test at the 0.01 level, indicating a good correlation and validity between the variables.

Table 2 Cronbach's Alpha of Indicators of NEU

Variable Cronbach's alpha Variable Cronbach's alpha Code Cronbach's alpha

X1 0.816 X12 0.8293 NEU 0.838

X2 0.8332 X13 0.8302

X3 0.828 X14 0.8181

X4 0.8266 X15 0.8319

X5 0.8428 X16 0.8339

X6 0.8253 X17 0.8314

X7 0.8395 X18 0.8357

X8 0.8334 X19 0.8266

X9 0.8377 X20 0.8467

X10 0.819 X21 0.8406

X11 0.8319 X22 0.8312

Table 3 KMO and Bartlett Test of Sphericity Results

Variable KMO Variable KMO Variable KMO Bartlett test of sphericity

X1 0.7854 X12 0.8509 Overall 0.7792 Chi-square 33184.396***

X2 0.5740 X13 0.8951

X3 0.8821 X14 0.8408

X4 0.7905 X15 0.8423

X5 0.6430 X16 0.8999

X6 0.8571 X17 0.7079

X7 0.6819 X18 0.8400

X8 0.6446 X19 0.7768

X9 0.4807 X20 0.6384

X10 0.8891 X21 0.6939

X11 0.8845 X22 0.7233

2. This study utilizes the relationship between residual plots and fitted values to evaluate the regression results, ensuring that the model meets the assumptions of normality, homoscedasticity, and independence. According to statistical theory, regression residuals should approximately follow a normal distribution with a mean of zero. If the residual distribution exhibits deviations from the zero line, curved patterns, or funnel shapes, it may indicate model misspecification or omission of important variables, potentially leading to biased regression results. Therefore, residual evaluation is conducted on the benchmark regression results of the impact of NEU on the "quantity increase" of GI (GIN) and the "quality improvement" of GI (GNS). The following analysis has been added to the "4.1 Benchmark Regression" section of the paper.

To further verify that the explanatory variable NEU affects GIN and GNS model meets the requirements of, normality, chi-square and independence, this paper evaluates the plots of fitted distributions of the residuals of the regressions of NEU on GIN and GNS. First, the fitted values and residuals are calculated for the regression results of NEU affecting GIN, and the plots of residual fitted values are plotted as shown in Figure 1. The scatter points are uniformly distributed on both sides of the reference line at y = 0. There is no systematic bias where all points lie above or below the reference line, indicating that the model is well unbiased and does not consistently overestimate or underestimate the observations. The scatter cloud is randomly distributed and does not form U-shaped, inverted U-shaped, or curved banding patterns, indicating that the main linear relationships of the data have been captured in the model and no important nonlinearities have been missed. The vertical range of fluctuations in the residuals is generally consistent over the vast majority of the range of fitted values (X-axis), with no curvilinear trends, funnel shapes, or other systematic patterns, a good indication of satisfying the assumption of homoscedasticity. Plotting the normal probability of the residuals (Q-Q plots) meets the linear regression model's requirement for normality of the residuals. Next, the fitted values and residuals are calculated for the regression results of NEU affecting GNS, and the residual fitted values are plotted, as shown in Figure 2. The scatter fluctuates roughly around and above the horizontal axis (residuals are 0) and there is no obvious pattern in the distribution of data points, indicating that the model is not systematically biased, i.e., it does not consistently overestimate or underestimate the actual values. The absence of hidden nonlinear patterns in the residuals indicates that the model has successfully captured the main linear relationships in the data. The range of fluctuations in the residuals is relatively stable over the entire range of fitted values, and the scatter cloud does not show a clear "funnel shape". Plotting the normal probability of the residuals (Q-Q plot) meets the requirement of residual normality for the linear regression model.

Figure 1 NEU Affects GIN's Residual Error vs. Fitted Value Graph（Please refer to the paper for details）

Figure 2 NEU Affects GNS's Residual Error vs. Fitted Value Graph（Please refer to the paper for details）

3. The original data used in this study have been uploaded to a public data platform, with detailed information provided in the "Data Availability Statement" section of the paper. Additionally, in order to further enhance data transparency and reproducibility, we have supplemented a file containing the raw data for the three-level indicators of the explanatory variable (New Urbanization, NEU) to the designated data repository.

The specific statement in the "Data Availability Statement" section of the paper is as follows:

DOI: 10.5281/zenodo.17547697

Cite as: Fang, L. (2025). The dataset for the paper titled "The impact of new urban-ization on quantity increase and quality improvement of green innovation: Based on multivariate moderating effect models" [Data set]. Zenodo. https://zenodo.org/records/17547697

Response to Reviewer #2 Comments

Reviewer #2:

This revised manuscript addresses a highly relevant and timely topic concerning the relationship between New Urbanization (NEU) and Green Innovation (GI) in the context of China's national development strategy. The study uses empirical analysis based on panel data from 284 cities spanning 2011 to 2022. The methodology appears strong, incorporating various econometric models, including fixed effects, robustness checks (removing special city/year samples, replacing variables), and instrumental variable methods (LAG_NEU and NLV) to address endogeneity concerns. The primary findings that NEU positively contributes to both the quantity increase (GIN) and quality improvement (GNS) of GI, and that Digital Inclusive Finance (DIF) significantly amplifies this impact offer significant policy implications and make a valuable contribution to the literature. Importantly, the authors are commended for making substantial revisions in response to the previous reviewers' comments, particularly regarding data transparency, discussion depth, and linguistic clarity. These revisions have significantly improved the quality and rigor of the manuscript.

Response:

Thank you very much for your positive and encouraging feedback on our revised manuscript. We are truly honored by your recognition of the study's relevance, methodological rigor, and substantive contributions. Your acknowledgment of the revisions made—particularly regarding data transparency, discussion depth, and linguistic clarity—is greatly appreciated and motivates us to continue striving for excellence in our research. We are glad that you find the empirical approach sound and the policy implications valuable. Your supportive comments reinforce our confidence in further exploring this important topic and pursuing rigorous scientific inquiry. Should any further refinements or clarifications be needed, we are more than willing to address them promptly.

---

## [Decision Letter · Decision Letter 2]

26 Feb 2026

PONE-D-25-37471R2The Impact of new urbanization on quantity increase and quality improvement of green innovation: Based on multivariate moderating effect modelsPLOS One

Dear Dr. Fang,

Thank you for submitting your manuscript to PLOS ONE. After careful consideration, we feel that it has merit but does not fully meet PLOS ONE’s publication criteria as it currently stands. Therefore, we invite you to submit a revised version of the manuscript that addresses the points raised during the review process.

We look forward to receiving your revised manuscript.

Kind regards,

Alejandro Botero Carvajal, Ph.D

Academic Editor

PLOS One

Journal Requirements:

Reviewers' comments:

Reviewer's Responses to Questions

**Comments to the Author**

1. If the authors have adequately addressed your comments raised in a previous round of review and you feel that this manuscript is now acceptable for publication, you may indicate that here to bypass the “Comments to the Author” section, enter your conflict of interest statement in the “Confidential to Editor” section, and submit your "Accept" recommendation.

Reviewer #4: All comments have been addressed

Reviewer #5: (No Response)

Reviewer #6: All comments have been addressed

Reviewer #7: All comments have been addressed

Reviewer #8: All comments have been addressed

2. Is the manuscript technically sound, and do the data support the conclusions?

Reviewer #4: Yes

Reviewer #5: Yes

Reviewer #6: Yes

Reviewer #7: Yes

Reviewer #8: Yes

3. Has the statistical analysis been performed appropriately and rigorously? 

Reviewer #4: Yes

Reviewer #5: Yes

Reviewer #6: Yes

Reviewer #7: Yes

Reviewer #8: Yes

4. Have the authors made all data underlying the findings in their manuscript fully available?

Reviewer #4: Yes

Reviewer #5: Yes

Reviewer #6: Yes

Reviewer #7: Yes

Reviewer #8: Yes

5. Is the manuscript presented in an intelligible fashion and written in standard English?

Reviewer #4: Yes

Reviewer #5: No

Reviewer #6: Yes

Reviewer #7: Yes

Reviewer #8: Yes

6. Review Comments to the Author

Reviewer #4: (No Response)

Reviewer #5: This manuscript empirically examines the impact of new urbanization on the quantity and quality of green innovation using panel data from Chinese cities and multivariate moderating models. The topic is relevant and potentially publishable; however, several areas require clarification or strengthening before acceptance.

1. While the paper clearly states its research objective, the conceptual links between NEU, GI, and moderating variables should be more rigorously theorized. The manuscript should explicitly justify the selection of proxies for GI (green patent applications and green invention patents), including potential measurement biases and alternatives.

2. Please provide variance inflation factors (VIF) or correlation matrices to improve credibility.

3. The manuscript reports significant coefficients and adjusted R² values but should provide fuller statistical interpretation. Consider reporting standardized coefficients to enhance interpretability.

4. Some interpretations overgeneralize findings beyond the empirical scope. For example, policy recommendations occasionally extend beyond what panel regressions alone can substantiate.

5. The manuscript is generally intelligible but contains frequent long and repetitive sentences that reduce readability. I would recommend professional language editing and tightening of phrasing in Results interpretation, Discussion, and Policy implication sections.

6. Some tables are dense and difficult to interpret; improved formatting or summary visualization could enhance accessibility.

7. Please consider expanding engagement with recent international literature beyond region-specific context to strengthen positioning within global research.

8. Highlight how the contribution advances theory rather than primarily confirming existing findings.

9. How sensitive are the results to alternative measures of green innovation or urbanization indicators and to different fixed-effects structures or clustering strategies?

10. What evidence supports the validity and strength of instruments used? Please Clarify.

11. The manuscript relies heavily on reduced-form econometric validation. Could the authors clarify how their findings advance causal inference rather than association-based policy recommendation?

12.What is the manuscript’s core methodological contribution beyond application of established panel techniques to a new dataset?

Reviewer #6: This paper examines whether new urbanization in China promotes both the quantity and the quality of green innovation using panel data from 284 cities during 2011–2022. The findings indicate that new urbanization significantly enhances green innovation, with digital inclusive finance and government-related factors further strengthening this positive effect. Please check the attached document for suggestions and recommendations.

Reviewer #7: (No Response)

Reviewer #8: Review report (date: 26.2.2026)

The Impact of new urbanization on quantity increase and quality improvement of green innovation: Based on multivariate moderating effect models, Manuscript Number: PONE-D-25-37471R2

General issues (taking into account seven guiding principles): Since I have viewed both the original and revised paper.

A1. The research is the original results of the study. The paper explores the linkage between new urbanisation and green innovation practices in Chinese cities, hence, presenting an example that can be used in policy and research in other nations and in filling the research gap in the field.

A2. There has been no publication of the results obtained. The manuscript utilises a micro-panel dataset spanning 2011 to 2022 (11 years) that makes the authors depict the time-related tendency in green innovation in the context of new urbanisation.

A3. Experiments, statistics, and other discussions are done to a high technical standard, and they are described in adequate detail. Basic pre-tests have been met, and the obtained estimates are sound, which makes it easy to replicate other researchers.

A4. The conclusions (summary) are made in a suitable way and are backed up by data. The conclusions are made logically out of the key findings and presented in a clear, scientific and concrete way.

A5. The article is written in the regular English. The paper is logically articulated and its structure is very well detailed.

A6. The study adheres to the necessary requirements regarding the ethics of research and experimentation. Following the examination of the document, critical remarks by reviewers and revision comments by authors were dealt with accordingly with the technical problems being fixed.

A7. The article follows the right reporting conventions and research ethics concerning data publication.

Hence, I would advise that the paper can be published in PLOS ONE journal.

7. PLOS authors have the option to publish the peer review history of their article (what does this mean?). If published, this will include your full peer review and any attached files.

Reviewer #4: No

Reviewer #5: No

Reviewer #6: No

Reviewer #7: No

Reviewer #8: **Yes:**Gedefaw Abebe Abiye

---

## [Author Response · Author response to Decision Letter 3]

6 Apr 2026

Response to Reviewers’ Comments

(Manuscript Number: PONE-D-25-37471R2)

Dear Reviewers,

We are very grateful for the reviewers’ constructive comments and suggestions on our manuscript entitled “The impact of new urbanization on "quantity increase and quality improvement" of urban green innovation” (ID: PONE-D-25-37471R2). Below, we provide a point-by-point response to each of the reviewers' concerns:

Response to Reviewer #5 Comments

Reviewer #5:

This manuscript empirically examines the impact of new urbanization on the quantity and quality of green innovation using panel data from Chinese cities and multivariate moderating models. The topic is relevant and potentially publishable; however, several areas require clarification or strengthening before acceptance.

Comment 1:

1. While the paper clearly states its research objective, the conceptual links between NEU, GI, and moderating variables should be more rigorously theorized. The manuscript should explicitly justify the selection of proxies for GI (green patent applications and green invention patents), including potential measurement biases and alternatives.

Response:

We are deeply grateful for the reviewer's insightful and constructive comments. These suggestions are pivotal for enhancing the theoretical depth and methodological rigor of our study. We have carefully addressed each point in the revised manuscript as detailed below.

We agree that a robust theoretical framework is the backbone of our research. In the revised manuscript (primarily in Theoretical analysis and research hypotheses sections), we have significantly expanded our theoretical exposition to build a rigorous logic chain connecting New Urbanization (NEU), Green Innovation (GI), and the five moderating variables.

First, we have further elaborated on the theoretical connection between the two concepts from the perspectives of New Urbanization (NEU) and Green Innovation (GI). We clarify that NEU transcends mere population agglomeration and spatial expansion; it represents a modern urban development paradigm characterized by economic growth, social coordination, and environmental sustainability. Green Innovation (GI), on the other hand, emphasizes the development and upgrading of green technologies aimed at resource recycling and environmental improvement. It encompasses creative activities across multiple dimensions, including technology, management, and institutions. From a conceptual logic standpoint, NEU constitutes the real-world context and demand engine for GI, while also providing an experimental space and an incentive framework for it, making it a key driving force behind Green Innovation. The paper analyzes how NEU facilitates the theoretical link between NEU and GI through aspects such as resource agglomeration, market demand, resource spatial transfer, and industrial integration. Furthermore, we have supplemented the theoretical logic of the relationship between NEU (New Urbanization) and GI (Green Innovation) by incorporating arguments from both Endogenous Growth Theory and the Constraint-Induced Innovation Mechanism.

The following specific content has been added to the first paragraph of the "Theoretical analysis and research hypotheses" section:

NEU goes beyond population concentration and spatial expansion, embodying a modern urban development paradigm characterized by economic growth, social harmony, and environmental sustainability. NEU represents a systematic reconfiguration of traditional urbanization, which is typically defined by “population mobility and urban expansion.” GI, on the other hand, emphasizes the development and upgrading of green technologies aimed at resource recycling and environmental improvement; it encompasses creative activities across multiple dimensions, including technology, management, and institutional frameworks. From a conceptual perspective, NEU constitutes the practical context and demand engine for GI, while also providing a testing ground and incentive framework for GI, serving as a key driving force behind it.

In the third paragraph of the "Theoretical analysis and research hypotheses" section, the following content has been specifically added:

Furthermore, endogenous growth theory holds that knowledge and technological information are the core drivers of sustained economic growth [25]. NEU brings together resources and factors within a specific geographical area to form a high-density, multi-stakeholder innovation network [26]. Various actors interact frequently within innovation networks; the industry-academia-research collaboration system formed within cities facilitates the rapid dissemination of knowledge and information, accelerating the transmission and recreation of technical knowledge, and leading to knowledge spillover effects and continuous innovation [27]. Constrained by the need for sustainable use of the environment and resources, companies are compelled to seek alternative technologies that meet environmental standards, thereby driving the innovation and development of green technologies.

Second, in the fourth paragraph of the "Theoretical analysis and research hypotheses" section, the following content has been specifically added:

Digital inclusive finance (DIF) is a form of finance that primarily relies on digital technology to provide convenient financial products and services to various segments of society[28]. NEU cannot function without strong financial support, DIF can provide convenient, low-cost financial services to various stakeholders in NEU, particularly rural migrant workers, enterprises, and service providers[29]。

Through the DIF platform, R&D enterprises can easily secure financing to sup-port GI activities—such as green technology R&D and the procurement of green equipment—thereby driving the production of green outcomes [33].

Third, in the fifth paragraph of the "Theoretical analysis and research hypotheses" section, the following content has been specifically added:

Environmental regulation theory posits that government intervention in pollution control promotes green innovation in the long run. Government technical support directly reduces the fixed costs and uncertainty associated with green R&D, increases firms’ willingness to invest in R&D in sectors such as new energy, energy conservation, and environmental protection, and ensures that knowledge spillovers generated during the NEU process are more concentrated in environmentally friendly technologies[40].

Informatization is the process of applying modern information technology to optimize resource allocation, improve operational efficiency, and drive economic restructuring[42]. Informatization can build online platforms and networks to break geographical restrictions, and has a positive effect on the equalization and popularization of public services[43]. Migrant workers can access convenient and efficient public services, which will help enhance the level of smart urban management and support the development of NEU.

Capital, technology, and talent are vital resources for economic development. Economically advanced regions typically possess greater financial resources for the research, development, and implementation of green technology infrastructure. They can increase investment in renewable energy technology and environmental conservation sectors to promote the improvement of urban infrastructure and the enhancement of public services, and promote the output of GI results [46].

Urbanization requires large quantities of energy-intensive products such as steel and cement. As rural migrants move into urban areas, their energy consumption primarily consists of commercial energy sources such as electricity and natural gas, leading to an increase in energy consumption[47]. In general, high-energy-consuming cities tend to face greater pressure on energy consumption and environmental pollution [48], and this compels cities to prioritize GI in the process of NEU.

Fourth, we first acknowledge that the measurement of green innovation (GI) relies on established classification systems based on predefined green innovation lists. Although these are the most widely accepted standards, they may still be subject to bias or fail to capture all green innovation outcomes. Our choice of green patent applications and green invention patents is primarily based on the fact that these two aspects directly align with the core concept of green innovation (GI). This is because green patent data provides a concrete and tangible measure of innovative activity; the number of applications reflects both the effort and intent behind innovation, thus offering a reasonable way to assess both the process and outcomes of green innovation. This approach is consistent with mainstream literature on environmental and ecological innovation (e.g., Pan, X. et al., 2021), which has confirmed that green patent applications and green invention patents serve as reliable proxy variables for green innovation. Furthermore, robustness checks using alternative measures (see Section 4.2 Robustness test) yielded stable results. In the robustness test, the total number of authorized green patents per 10,000 people (NGIN) was used to measure the “incremental” aspect of green innovation, while the total number of authorized green invention patents per 10,000 people (NGNS) was used to measure the “quality improvement” aspect of green innovation.

The following specific content has been added to the "3.1. Indicator design" section of the paper：

The total amount of green patent applications not only reflects the situation of innovation activities, but also is closely related to the economic and environmental benefits of GI. Green patent data provides a specific and tangible result measurement standard for innovation activities. The number of applications reflects the innovation efforts and intentions. The number of green patent applications can not only better measure the process of GI, but also directly reflect the research and development results of enterprises, scientific research institutions or individuals in the field of green technology. The number of green patent applica-tions is standardized, quantifiable and internationally universal, which is convenient for statistical analysis and horizontal comparison. This choice is consistent with the mainstream literature on environmental innovation and ecological innovation [51,52].

Comment 2:

2. Please provide variance inflation factors (VIF) or correlation matrices to improve credibility.

Response:

To further examine multicollinearity from a statistical perspective, this study calculated the Variance Inflation Factor (VIF) values between the independent variable NEU and the control variables FDI, INT, RFV, HCL, OOW, and SCL. The VIF values for all variables were found to be less than 5, indicating that there is no significant multicollinearity issue between NEU and the control variables. To clearly present the results of the multicollinearity test, we have added Table 5 in the paper to display the VIF data, and included the following content at the end of the "3.3. Data sources and descriptive statistics" section:

In order to further test the multicollinearity, this paper calculates the VIF ( variance inflation factor ) values of the independent variable NEU and the control variables FDI, INT, RFV, HCL, OOW, SCL. As shown in table 5, the VIF values of all variables are less than 5, and the correlation coefficients between each variable are also less than 0.7. It is considered that there is no multicollinearity problem between NEU and control variables.

Table 5 The variance inflation factor and correlation coefficient of each variable

NEU FDI INT RFV HCL OOW SCL

Variance Inflation Factor VIF 2.94 1.17 1.27 1.01 1.92 1.55 1.04

1/VIF 0.3398 0.8536 0.7876 0.9881 0.5211 0.6452 0.9572

correlation coefficient NEU 1

FDI 0.2341 1

TEL 0.5943 0.3153

INT 0.4203 -0.0321 1

RFV 0.0176 -0.0254 0.0225 1

HCL 0.6714 0.2274 0.3228 0.0117 1

OOW 0.5489 0.2205 0.2021 -0.0250 0.2783 1

SCL 0.0850 -0.0203 0.0772 -0.0814 0.1467 -0.0531 1

Comment 3:

3. The manuscript reports significant coefficients and adjusted R² values but should provide fuller statistical interpretation. Consider reporting standardized coefficients to enhance interpretability.

Response:

(1) We sincerely thank the reviewer for the constructive comment. In the revised manuscript, we have expanded the depth of interpretation for the statistical results. In the "4.1 Benchmark regression" section, we explicitly state the impact coefficient of NEU and its significance level, provide a specific explanation of the economic meaning of this coefficient, and elaborate in detail on its practical implications. Specifically, we have added the following content:

This means that for every 1 unit increase in NEU, GIN increases by an average of 0.7452 units, GNS increases by an average of 0.7892 units, the adj.R2 in E1 and E3 increases from 0.462 to 0.861, and the adj.R2 in E2 and E4 increases from 0.344 to 0.849. The overall fitting degree of the model is improved, and the control variables such as FDI, INT, RFV, HCL, OOW and SCL in the model explain the 'residual variation', which leads to the improvement of the overall fitting degree of the model. This further verifies that NEU is an important driving force for GIN and GNS, and also shows that the model effectively captures more comprehensive driving factors.

(2) Regarding the "4.4. Moderating effects" section, we have supplemented our analysis by discussing the coefficients, significance levels, and changes in the adjusted R², along with explaining their practical implications. We also added an interpretation of the adjusted R² value, clarifying how changes in the adjusted R² affect the explanatory power of the model. By incorporating p-values, we explained the practical meaning of statistical significance.

Specifically, we have added the following content:

The adj. R2 in E23 and E24 are 0.903 and 0.891, respectively. The two models fit well. DIF has a complementary moderating effect, and the moderating mechanism of DIF is strongly supported by the sample data. For every unit increase in the moderating variable DIF, the marginal effect of NEU on GIN increases by 0.0025 units, and the marginal effect of NEU on GNS increases by 0.0026 units.

The adj. R2 in E25 and E26 are 0.905 and 0.790, respectively. The two models fit well. DIF has a complementary moderating effect. In strong government support cities, for every unit increase in the moderating variable DIF, the marginal effect of NEU on GIN increases by 0.0020 units. In weak government support cities, it increases by 0.0015 units.

The adj. R2 in E29 and E30 are 0.921 and 0.882, respectively. The two models fit well. DIF has a complementary moderating effect. In high-information cities, for every unit increase in the moderating variable DIF, the marginal effect of NEU on GIN increases by 0.0018 units, while in low-information cities, it increases by 0.0015 units.

The specific content can be found in the marked sections of the manuscript.

(3) To enhance the interpretability of the model results, we have standardized the main explanatory variable NEU as well as the dependent variables GIN and GNS, following your suggestion. In the regression analysis based on the standardized variables, we report the standardized regression coefficients, which directly reflect the number of standard deviations the dependent variable changes for each one-standard-deviation change in the independent variable. This facilitates comparison of the relative impact of different variables on the outcome. The regression coefficients in Tables 6–14 have been updated to reflect the standardized regression results, and the modified parts have been marked in the paper.

Comment 4:

4. Some interpretations overgeneralize findings beyond the empirical scope. For example, policy recommendations occasionally extend beyond what panel regressions alone can substantiate.

Response:

We fully agree with your opinion. Upon careful reflection, we realized that when drafting the Discussion and Recommendations section, we did indeed overanalyze the "association findings based on panel data," and some of the proposed policy recommendations wer

---

## [Decision Letter · Decision Letter 3]

1 May 2026

The impact of new urbanization on "quantity increase and quality improvement" of urban green innovation

PONE-D-25-37471R3

Dear Dr. Fang,

We’re pleased to inform you that your manuscript has been judged scientifically suitable for publication and will be formally accepted for publication once it meets all outstanding technical requirements.

Kind regards,

Alejandro Botero Carvajal, Ph.D

Academic Editor

PLOS One

Additional Editor Comments (optional):

Reviewers' comments:

Reviewer's Responses to Questions

**Comments to the Author**

1. If the authors have adequately addressed your comments raised in a previous round of review and you feel that this manuscript is now acceptable for publication, you may indicate that here to bypass the “Comments to the Author” section, enter your conflict of interest statement in the “Confidential to Editor” section, and submit your "Accept" recommendation.

Reviewer #5: (No Response)

Reviewer #6: All comments have been addressed

2. Is the manuscript technically sound, and do the data support the conclusions?

Reviewer #5: Yes

Reviewer #6: Yes

3. Has the statistical analysis been performed appropriately and rigorously? 

Reviewer #5: Yes

Reviewer #6: Yes

4. Have the authors made all data underlying the findings in their manuscript fully available?

Reviewer #5: Yes

Reviewer #6: Yes

5. Is the manuscript presented in an intelligible fashion and written in standard English?

Reviewer #5: No

Reviewer #6: Yes

6. Review Comments to the Author

Reviewer #5: 1. While the authors have appropriately toned down overgeneralizations, a few sentences still imply stronger causality than the observational design warrants. For example: "identify a causal effect of NEU", given that "Methods" address endogeneity but still rely on exclusion restriction assumptions, consider softening to "suggest a causal relationship" or "provide evidence consistent with a causal effect."

2. The authors abandoned GMM due to overidentification concerns which is a valid point and instead used a lagged dependent variable in a fixed effects framework. However, the "dynamic panel bias" in short-T panels should be acknowledged. With T=12, the bias may be modest but mention should be made in limitations.

3. Despite professional editing, a few sentences remain unclear and several instances of inconsistent spacing around em dashes and parentheses. One final light proofread would eliminate these residual issues.

4. The energy consumption variable (ENC) should be clarified

5.The conclusion would benefit from a more explicit acknowledgment that the findings are specific to China's institutional context, with suggestions for which aspects might generalize to other emerging economies.

6.Add a short paragraph in Section 5 interpreting why the relationship is characterized by a single structural break rather than multiple regimes. This would turn a purely statistical result into a more insightful finding.

Reviewer #6: The authors have revised the manuscript carefully and made significant improvements to its overall quality. The main issues raised by the reviewers have been properly addressed, and the revised version is much clearer and stronger than before. I appreciate the authors’ efforts in improving the paper. In its current form, the manuscript is suitable for acceptance and publication.

7. PLOS authors have the option to publish the peer review history of their article (what does this mean?). If published, this will include your full peer review and any attached files.

Reviewer #5: No

Reviewer #6: No

---

## [Editor Report · Acceptance letter]

PONE-D-25-37471R3

PLOS One

Dear Dr. Fang,

I'm pleased to inform you that your manuscript has been deemed suitable for publication in PLOS One. Congratulations! Your manuscript is now being handed over to our production team.

Kind regards,

on behalf of

Dr. Alejandro Botero Carvajal

Academic Editor

PLOS One